# Embodied working memory during ongoing input streams

**Nareg Berberian** *, **Matt Ross, Sylvain Chartier**

Laboratory for Computational Neurodynamics and Cognition, School of Psychology, University of Ottawa, Ottawa, Ontario, Canada

* nberb062@uottawa.ca

## Abstract

Sensory stimuli endow animals with the ability to generate an internal representation. This representation can be maintained for a certain duration in the absence of previously elicited inputs. The reliance on an internal representation rather than purely on the basis of external stimuli is a hallmark feature of higher-order functions such as working memory. Patterns of neural activity produced in response to sensory inputs can continue long after the disappearance of previous inputs. Experimental and theoretical studies have largely invested in understanding how animals faithfully maintain sensory representations during ongoing reverberations of neural activity. However, these studies have focused on preassigned protocols of stimulus presentation, leaving out by default the possibility of exploring how the content of working memory interacts with ongoing input streams. Here, we study working memory using a network of spiking neurons with dynamic synapses subject to short-term and long-term synaptic plasticity. The formal model is embodied in a physical robot as a companion approach under which neuronal activity is directly linked to motor output. The artificial agent is used as a methodological tool for studying the formation of working memory capacity. To this end, we devise a keyboard listening framework to delineate the context under which working memory content is (1) refined, (2) overwritten or (3) resisted by ongoing new input streams. Ultimately, this study takes a neurorobotic perspective to resurface the long-standing implication of working memory in flexible cognition.

## Introduction

Animals are capable of relying on internal sensory representations rather than purely on the basis of external stimuli. These representations routinely support higher-order functions such as working memory (WM)—holding the stimulus content in memory for a certain duration in the absence of its concrete presence [1–6]. Experimental studies on WM in primate neurophysiology have shown that patterns of neural activity can reverberate following the offset of sensory inputs [4, 7]. These ongoing reverberations can persist over a period of several seconds [1, 5–8], a time segment during which neuronal responses remain preferentially elevated for a target stimulus [9–11]. Despite having shown to be correlated with psychophysical performance [7], mechanisms underlying intrinsic dynamics of self-sustained activity remain elusive.

**Data Availability Statement:** The data and code underlying this study are available on the Open Science Framework (DOI: 10.17605/OSF.IO/SXZTV, URL: https://osf.io/sxztv/).

**Funding:** This research was supported by the Natural Sciences and Engineering Research

Council of Canada (NSERC) and the Ontario Graduate Scholarship (OGS) Award Program. The funders had no role in study design, data collection and analysis, decision to publish, or preparation of the manuscript. First Author, Nareg Berberian received OGS funding for this work. Second Author, Matt Ross did not receive funding for this work. Senior Author, Sylvain Chartier received NSERC funding for this work.

**Competing interests:** The authors have declared that no competing interests exist.

Consequently, the significance of reverberating activity for higher-order function remains unknown.

A long-standing question shared across theoretical and experimental studies of WM is how animals maintain a faithful representation of sensory stimuli during ongoing reverberations of neural activity. Persistent activity during WM has been studied extensively via influential observations of experimental data [8, 12, 13]. These experiments have motivated the development of computational models, which in turn have triggered fundamental theoretical insights into potential mechanisms underlying WM capacity [14–19]. These formal models, among others, form recurrently connected networks, which typically maintain selective elevated persistent activity through local excitatory recurrent connections with global feedback inhibition [20]. In this formalism, sustained firing may be achieved by carefully fine-tuning the strength and structure of recurrent circuitry [21]. As such, these functional networks can maintained, in memory, the spatial location of target stimuli, forming what is commonly known as persistent activity bumps (i.e. bump attractors) [20].

Persistent activity in recurrent networks has been supported by short-term synaptic plasticity, where synaptic strength is rapidly regulated by recent historical activity within the network [16, 22, 23]. In the presence of such rapid changes in synaptic dynamics, neuronal activity can drift over time [16, 22], or remain centered at the initial bump location [23]. Despite their overarching support from empirical studies [24, 25], WM models holding the persistent memory hypothesis have for long been using preassigned protocols of fixed stimulus presentation. In this context, previous methods have primarily focused on one aspect of WM, namely the maintenance of a specific target stimulus—which leaves out by default its interaction with ongoing input streams. In a real-world setting, biological agents are continuously bombarded with stimuli, some of which they maintain in WM while new input streams actively harness their ongoing sensory experience. How does the brain manage to maintain specific target stimuli in WM, while withstanding streams of incoming stimuli? As the environment continuously delivers a rich repertoire of sensory intricacies, the brain must somehow reconcile these two seemingly contradictory tasks—maintaining previous inputs stable over a period of time while new input streams are continuously coming in. To this end, WM must have a temporal undercurrent, a basis under which it integrates past and present time points.

In contrast to displaying persistent activity by recurrent interactions, here we instead devise a feedforward network of spiking neurons [26], subject to both short-term and long-term synaptic plasticity [27–30]. By considering activity-dependent Hebbian plasticity as a complementary mechanism for generating persistent activity [31], we examine how the content of WM interacts with the intricacies of ongoing input streams. Spiking networks are well-known for carrying out continuous online operations in non-stationary environments [32]. In this context, synaptic connections can be continuously modified [33], without fine-tuning their strength and structure. This ongoing process of modification may result in momentary network restructuring so as to accommodate an evolving environment [32]. Here, we present six different experiments, namely (1) single learning and recall, (2) incremental learning and recall, (3) task-switching, (4) resistance to interference, (5) submission to interference, and (6) resistance to distraction. Each experiment provides an entry point for the next one (see Robotic experiments), highlighting the scenario under which the WM content may be (1) refined, (2) overwritten or (3) resisted by the arrival of ongoing input streams.

Online unsupervised learning of WM capacity has been studied in networks of spiking neurons connected by plastic synapses (see [30] for review). These networks can undergo an ongoing process of structural modification, with no *a priori* knowledge about the content of subsequent input streams [30, 32, 34, 35]. In this context, previous target inputs may be gracefully overwritten by subsequent inputs, because synaptic plasticity can provide the malleability

needed for the content of WM to ride the wave of new input streams. Networks that operate in this fashion share the palimpsest property, meaning they forget the content of previous inputs to make room for new ones [36–38]. Under some circumstances however, the content of WM may still be retrieved despite the arrival of subsequent inputs, because plastic synapses behave in a context-specific manner [36]. For example, previous studies have shown that the stimulus statistics can have an impact on the state transition probability of the synapse—the probability of transitioning to a new state different from the previous state [30, 34].

The WM of biological agents operating in the real-world is inevitably carried out while subsequent input streams continuously arrive. In support of their biological counterparts, artificial agents require adaptive capabilities in order to be deemed useful in the presence of ongoing changes in the external world. Indeed, physical robots pose as ideal companions for carrying out sequential tasks online, whereby the artificial agent must process incoming information on-the-fly. Here, we embed the spiking network in the "Vector" robot produced by Anki developer. We perform our experiments using the agent as a tool for studying WM capacity during ongoing input streams. Importantly, the robot is used to ensure a one-to-one correspondence between network activity and behaviour—a companion tool for moving closer to the real-world interaction that exists between WM and ongoing input streams.

## Materials and methods

### Network architecture

The spiking neural network consisted of 500 units linked reciprocally. The initial network connectivity followed a set of rules [39]. First, inhibitory neurons occupy approximately 20% of the population of cortical neurons, whereas the great majority of remaining neurons are excitatory [40]. As such, 80% of units in the embodied network were randomly chosen to be excitatory, whereas the remaining 20% were inhibitory. Following Dale's law, a given excitatory/inhibitory unit only exhibited excitatory/inhibitory efferent connections, respectively. Second, the number of afferent synapses to a single neuron in cortex is limited, forming clusters of sparsely connected networks with roughly 10-20% probability of synaptic contact [41]. To this end, initial network connectivity was sparse, with only 20% of all possible connections present (chosen randomly among all possible connections). At network initialization, self-connections were not permitted because although not uncommon, they are rare *in vivo*, in comparison to their prominence in dissociated cell cultures [42]. Finally, efferent synaptic connections from excitatory units ($J_{EE}$, $J_{IE}$) exhibited potentiated efficacy, whereas those from inhibitory units ($J_{II}$, $J_{EI}$) were depressed. Network parameter values are indicated in Table 1.

**Table 1. Network parameters.**

| Symbol | Description | Value |
|---|---|---|
| $c$ | Initial probability of synaptic contact | 0.2 |
| $N_E$ | Number of E units | 400 |
| $N_I$ | Number of I units | 100 |
| $J_{EE}$ | Initial E to E synaptic efficacy | 0.65 |
| $J_{IE}$ | Initial E to I synaptic efficacy | 0.65 |
| $J_{II}$ | Initial I to I synaptic efficacy | −1 |
| $J_{EI}$ | Initial I to E synaptic efficacy | −1 |

## Robotic platform

The Vector Software Development Kit (SDK) was used as an open-source robotics platform. In particular, we used a physical robot named "Vector" by Anki Developer Fig 1. This robot is equipped with five basic hardware components, namely (1) an HD camera, (2) an infrared laser sensor, (3) a beamforming 4-microphone array, (4) a capacitive touch sensor and (5) two wheel motors. The computational resources of the robot were remote, performed on a laptop computer. Incoming stimuli were controlled via user keypresses from the laptop computer, and the communication was mediated via wireless network. This embedded control design was intended so as to avoid using Vector's onboard tactile sensors. To explain, the robot continuously moved during the experiments. Therefore, reliance on tactile sensors would demand the user to continuously follow the robot throughout the experiment. More-over, sensor reading is noisy and therefore some tactile triggers would go unnoticed from Vector failing to read them as valid touch. These would vastly limit the fluidity of informa-tion exchange, because they would place high workload demands on the user (but see [43]). Consequently, the time required for conveying user desirability would increase. For these reasons, we have instead devised a keyboard listening framework. The protocol was designed to ensure an efficient and instant medium of communication between the user and the robot [44]. Finally, there were no human subjects recruited in this study. The degree of user involvement in the teleoperation is similar to the degree of involvement in pressing a key to compile a code on a computer.

In order to grant access to the robot's hardware capabilities, API commands were commu-nicated to the corresponding actuators of the robot. The only hardware components used in this study were Vector's two wheel motors. The execution of ongoing motor trajectories was mediated by the spiking activity of the network. Spike count information was computed in consecutive non-overlapping time bins of 40ms. The rotating speed of the two wheel motors were decoded separately. Each motor rotated at a given speed (in millimetres per second) until the total spike count of the next non-overlapping time bin commanded the robot to drive at a new speed. Taken together, neuronal responses were mapped onto the wheel motors of the robot, controlling the execution of its navigated trajectory.

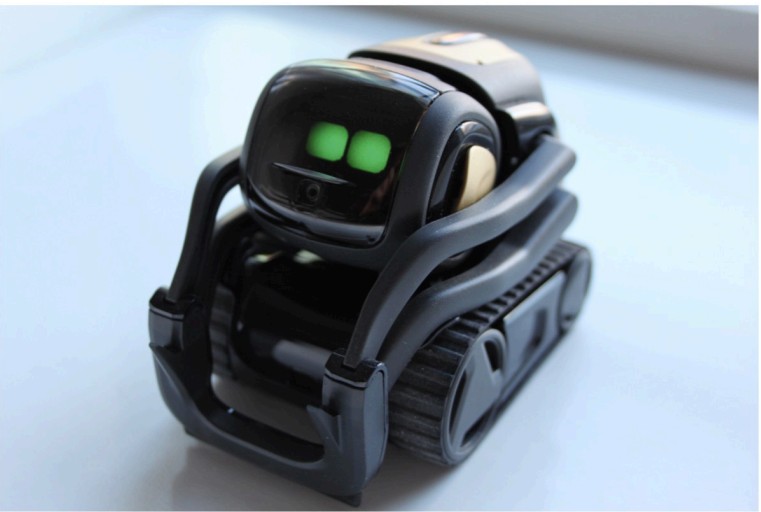

**Fig 1. Vector the robot.** Vector behaves as a result of network spiking activity.

## Sensory inputs

Sensory-evoked inputs were presented to the network in the form of spatially localized currents represented by a bimodal mixture of Gaussians:

$$I_i^{ext}(t) = R_b + R_p e^{-\frac{(i-i_{pref})^2}{2\sigma^2}} + R_n e^{-\frac{\left(i+\left(\frac{N}{2}\right)-i_{pref}\right)^2}{2\sigma^2}} \qquad (1)$$

where $R_b$ was the baseline amplitude of sensory inputs, $R_p$ was the higher amplitude peak whereas $R_n$ was the lower amplitude peak, $\sigma$ was the tuning width controlling the input specificity, $i$ was the unit index (1, 2, 3, . . .$N$) with $N$ corresponding to the number of units.

There are three main reasons why bimodal activations were used, including the specified offset between the two Gaussians. First, previous models of spatial WM have used the Gaussian for characterizing their target stimulus. Inputs range from a single target to multiple targets [14–16, 19, 23, 25, 45–50]. Second, Vector is equipped with two wheel motors. The agent was therefore an ideal candidate for supplying its differential steering system with bimodal activations. The specific offset between the Gaussians induced differences in wheel motor speed, which in turn resulted in rotating movements with diameters small enough for the robot to accommodate the confined spatial setting. Finally, the offset between the two Gaussians has experimental grounding, inspired from a variant of the classic oculomotor delayed response task [7]. In these experiments, animals are shown two simultaneously presented stimuli, with differences in luminance (i.e. contrast ratio).

The network involved two subpopulations, consisting of spiking units spatially distributed according to the sensory input to which they were most sensitive (Fig 2). The bimodal activations differentially drove the activity of units within each respective subpopulation, producing localized activity bumps determined by the spatial structure of injected external input currents $I_i^{ext}$. In particular, $i_{pref}$ corresponded to the unit most sensitive to $R_p$, and therefore exhibited the highest response amongst its neighbouring units within the subpopulation.

In this study, a keyboard listening framework was devised in order to actively control ongoing input streams to the embodied network (see Robotic platform). By delivering keypresses from a laptop computer, the user could control both the configuration and the duration of stimuli. Sensory inputs were represented by two distinct configurations (Fig 2). Their average intensity were identical. The only distinguishing characteristic between the two was their

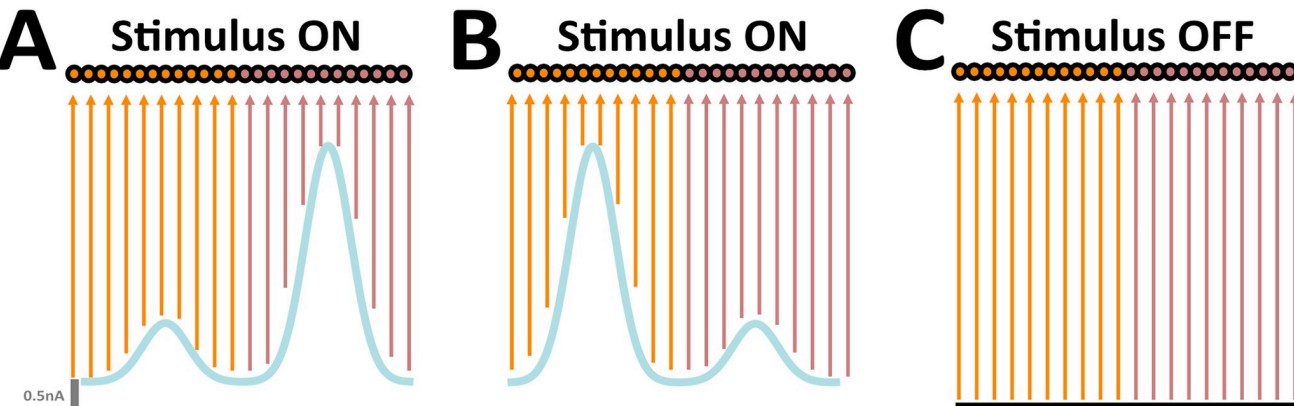

**Fig 2. Sensory afferents to the embodied network.** (A) ON stimulus (blue). (B) Same as (A), but instead peaks are interchanged (C) OFF stimulus (black); orange, subpopulation 1; brown, subpopulation 2. Parameters: $R_b$ = 0.5 nA, $R_p$ = 2.5 nA, $R_n$ = 1.0 nA, $i_{pref}$ = 125 for clockwise rotations, $i_{pref}$ = 375 for counter-clockwise rotations, $N$ = 500, $\sigma$ = 35. Figure adapted from [23].

interchangeable amplitude peaks (Fig 2A and 2B). In configuration 1, units in a given subpopulation received inputs near $R_p$, whereas those in the remaining subpopulation received inputs near $R_n$ (Fig 2A). In configuration 2, spatially localized amplitude peaks were interchanged (Fig 2B). Overall, inputs evoked localized activity profiles in the network (i.e. bump states). In their absence, $I_i^{ext} = 0$ (Fig 2C).

### Learning and recall procedure

During WM tasks (e.g. oculomotor delayed response), the animal experiences a "cue period" where external stimuli are presented. Stimulus presentation is immediately followed by a "delay period" where external stimuli are removed. As an analogy to typical experiments, our study treated the cue period as learning, and the delay period as recall. For simplicity, the "response period" normally characterized by saccadic eye movements towards a target location were not considered. Overall, the temporal evolution of the model was organized in batches of learning and recall in order to maintain an operational definition used by traditional approaches in artificial neural networks (ANNs).

Learning and recall phases were carried out by distinct time segments of network activity during which the robot exploited input-dependent and input-disengaged operations, respectively (Fig 3). The offset of both learning and recall is controlled by pressing the <Enter> key

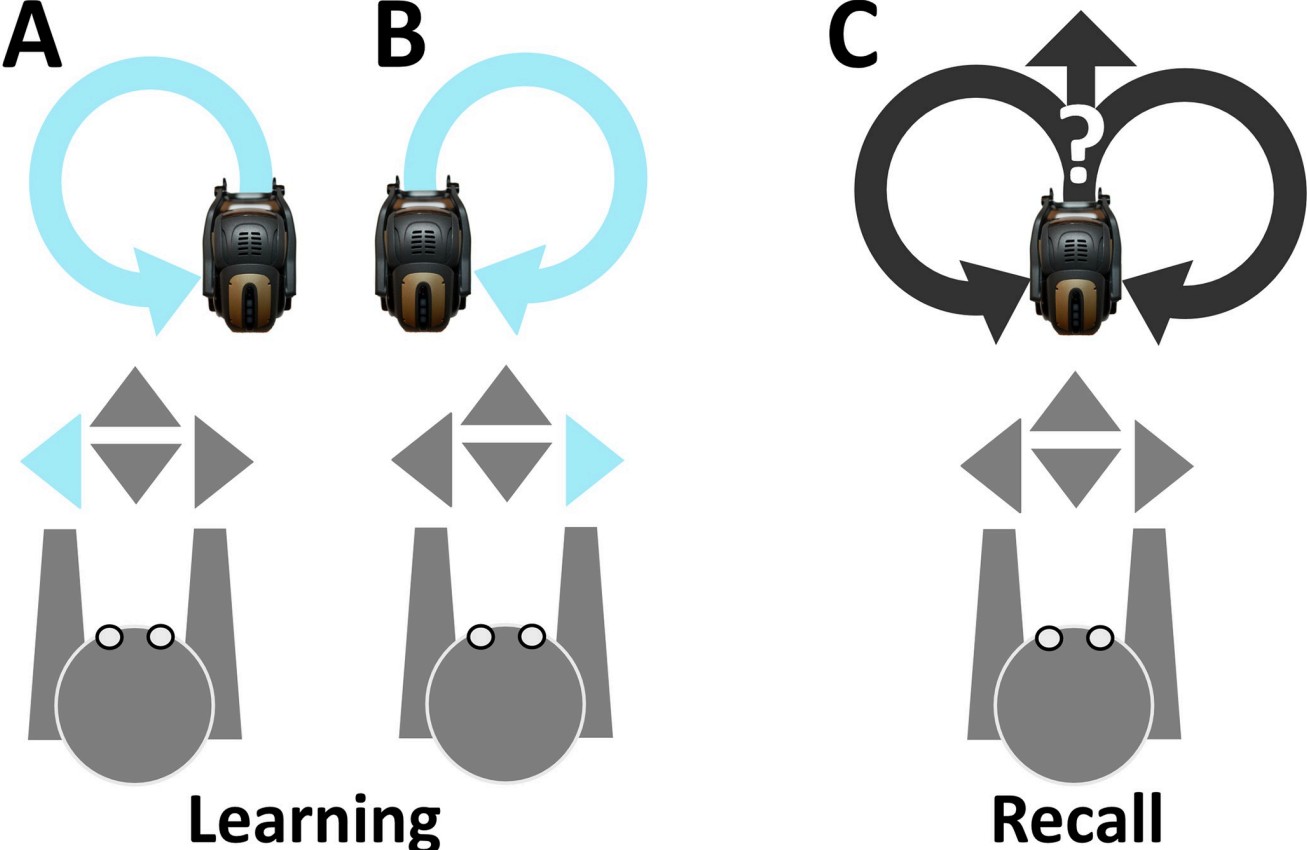

**Fig 3. Keyboard listening setup.** (A) The <Left> keypress recruits input configuration shown in Fig 2A, marking the onset of input-dependent counter-clockwise motor executions. (B) Same as in (A), but instead the <Right> key is pressed, recruiting input configuration shown in Fig 2B, marking the onset of input-dependent clockwise motor trajectories. (C) The <Enter> key is pressed (not shown), recruiting the homogeneous input configuration shown in Fig 2C, marking the onset of input-disengaged operations.

(at any time instance). To mark the onset of learning, the <Left> or <Right> key is pressed, such that the robot executes counter-clockwise or clockwise motor trajectories, respectively (Fig 3A and 3B). To end the learning phase, the <Enter> key is pressed (not shown), allowing the robot to move autonomously (Fig 3C). Overall, the learning phase was considered as a time segment of evoked activity where units were driven by sensory afferents. In contrast, the recall phase was considered as a time segment of spontaneous activity where units were solely driven by the intrinsic dynamics of the network.

Learning and recall phases were alternated by turning the stimulus ON and OFF, repeatedly (ON-OFF-ON-OFF…). This sequence-based approach allowed us to include but also move beyond traditional approaches of single learning and recall. The spiking network was subject to continuous modification. This ongoing process naturally removes the fine line that has traditionally been used to distinguish learning from recall in ANNs [33]. Activity-dependent changes in synaptic efficacy were exclusively based on local information (see Plasticity model). Evoked activity led to the formation of synaptic structure in a rate-dependent Hebbian manner [30]. The synaptic structure provided sufficient structured feedback so as to endow units with the ability to maintain stimulus-selective persistent activity [35, 51]. The structure of the weight matrix representing the stimulus information was put in a state of reverberation that outlasted the exposure of the stimulus. Hence, the role of the resulting synaptic structure was to sustain a local activity bump produced by previous stimuli. In this way, items that were loaded as discrete "slots" during learning were maintained by distinct localized activity during recall. If the stimulus was absent for long durations, network activity progressively decayed back to baseline levels [26]—an expression reflected in the network structure. Taken together, the initial Network architecture was not designed to meet the requirements for a particular task, but was rather subject to ongoing modification through the sensory experience of the robot [43].

## Robotic experiments

We report the procedure of six robotic experiments and lay out the criteria for their successful completion. In the first experiment, a single learning and recall procedure is introduced. Since network activity is unavailable to the user, the behaviour of the robot is monitored to gain insight into network convergence. Here, the criteria for successful task completion is to observe consistent movement trajectories from the robot. This approach differs from disembodied networks imprinted with a minimum stopping rule for weight convergence, because robotic behaviour is the marker of network stability, not preassigned internal triggers. Using movement trajectories as top-down evidence of network convergence, the user can then remove the stimulus and observe the recall performance of the robot. This observation is used as a stepping stone for introducing an incremental paradigm.

In the second experiment, an incremental learning and recall procedure is presented. This experiment encompasses the notion of behavioural shaping [52]. Here, shaping refers to an incremental process where the user provides feedback to an agent so as to improve approximations of a target behaviour [53, 54]. The user observes the behaviour of the robot in response to different feedback signals (e.g. stimulus duration), and makes the necessary adjustments on-the-fly. In this way, robotic behaviour is progressively fine-tuned, moving the agent closer to the desired behaviour [55]. This incremental process towards a single target behaviour begs the question, however, whether the robot can move beyond single task demands, by transitioning between multiple tasks.

In the third experiment, a task-switching procedure is introduced. The robot is shown two interchanging inputs. The criteria for success is the expression of flexible behaviour. Here,

sustained learning is not a necessary requirement for successful task completion, because task-switching washes away the memory of the previous task. The robot must adapt to change at every step of the way, accommodating user desirability. Nevertheless, is the robot always deemed to be accurately aligned with user judgements?

In the fourth experiment, a task interference procedure is introduced. Here, the robot is presented with a long cue stimulus, and tested against a brief interfering stimulus. To successfully accomplish this experiment, the robot must keep performing movement trajectories in the same direction as those observed during the cue period, even after the user interferes by imposing movements in the opposite direction. In this scenario, if the robot resists interfering inputs, is it because they are presented for a shorter duration or because the content of the cue stimulus is still maintained in WM?

In the fifth experiment, a variant of the task interference procedure is presented. Here, delay duration is extended long enough to wash-away the content of the cue stimulus. When the robot forgets the cue stimulus, its movement trajectory provides the user with enough evidence to present the interfering stimulus. To successfully perform this experiment, brief interfering stimuli should be enough to overwrite the content of cue stimuli. Otherwise, the previous experiment would suggest that the robot resisted interfering inputs simply because they were presented for a shorter duration, and not because the content of cue stimuli were maintained in WM. Nevertheless, is the robot capable of resisting distractions that last as long as the cue stimulus?

In the sixth and final experiment, the robot is tested against distractor inputs. Here, the duration of both cue and distractor inputs are similar. However, distractor intensity is three times weaker than cue intensity. The magnitude of distractor intensity is based on two requirements, namely (1) to impose distracting movements in a direction opposite to those imposed during the cue period and (2) to navigate within the confined spatial environment. To successfully accomplish this experiment, the robot must move in the same direction before and after distraction, despite movements in the opposite direction during distraction. Taken together, the behaviour of the robot, not just the network activity, is a requirement for the successful completion of our experiments.

## Spiking neural model

The spiking neural network was numerically simulated via current-based synapses. The network was composed of adaptive exponential integrate-and-fire (aEIF) units, a well-known model that captures the spike initiation behaviour and statistics of cortical neurons [56, 57]. Neuronal dynamics were described by a membrane potential $V_i$ and by a spike-frequency adaptation $w_i$—a widespread neurobiological phenomenon [58]. Below a constant threshold $V_T$, units evolved according to:

$$C_m \frac{dV_i}{dt} = -g_L(V_i - E_L) + g_L \Delta_T e^{\frac{V_i - V_T}{\Delta_T}} - w_i + I_i^{syn} + I_i^{ext} \quad (2)$$

where $C_m$ was the membrane capacitance, $g_L$ was the leak conductance, $E_L$ was the resting potential, and $\Delta_T$ was a slope factor, which determined the sharpness of the threshold. Here, $I_i^{syn}$ described the postsynaptic current generated from the intrinsic dynamics of the network, $I_i^{ext}$ was the sensory input for which its presence $I_i^{ext} > 0$ or absence $I_i^{ext} = 0$ was controlled using keypresses (see Sensory inputs). The postsynaptic membrane voltage (Eq 2) was coupled to an adaptation variable $w_i$ defined by:

$$\tau_w \frac{dw_i}{dt} = a_w(V_i - E_L) - w_i \quad (3)$$

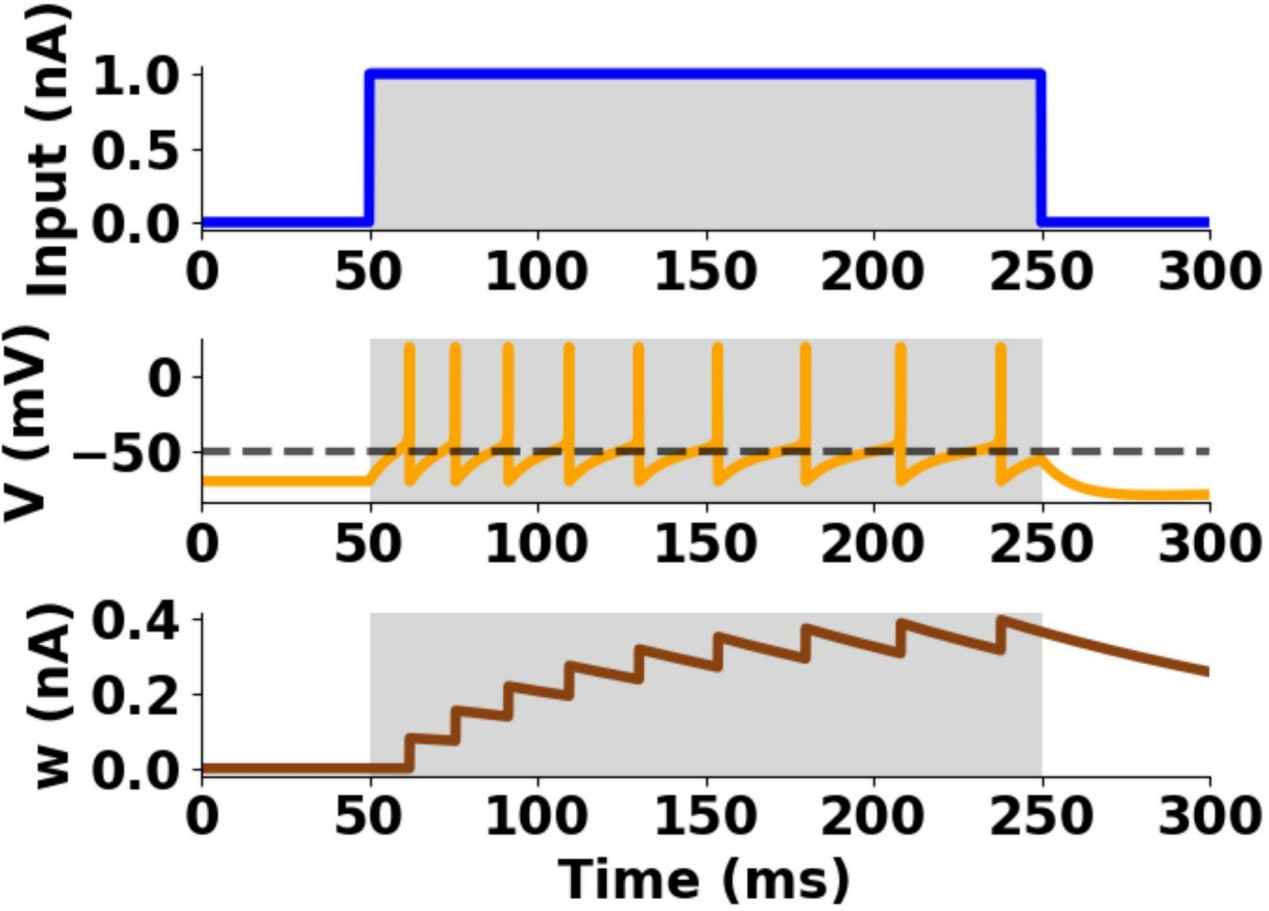

**Fig 4. Single unit profile of the aEIF model.** Postsynaptic spikes from input current injection. Top: input current, Middle: membrane potential, Bottom: adaptation variable. Parameters: $C_m$ = 281 pF, $g_L$ = 30 nS, $E_L$ = −70.6 mV, $V_T$ = −50.4 mV, $V_{peak}$ = 20 mV, $\Delta_T$ = 2 mV, $\tau_w$ = 144 ms, $a_w$ = 4 nS, $b_w$ = 0.0805 nA, $\tau_{syn}$ = 5 ms.

where $\tau_w$ was the adaptation time constant and $a_w$ represented the level of subthreshold adaptation. When the membrane potential crossed the threshold ($V_i > V_T$), a spike ($V_{peak}$) was triggered in the aEIF model, and the integration of Eq 2 was reset to $V_{reset}$ ($V_i \rightarrow V_{reset}$), with $V_{reset}$ = $E_L$. In parallel, the adaptation variable in Eq 3 was increased by an amount $b_w$ ($w_i \rightarrow w_i + b_w$), where $b_w$ was the parameter of spike-triggered adaptation. Single unit parameter values of the aEIF model were taken from [56], and kept fixed throughout all robotic experiments (Fig 4).

## Plasticity model

Short-term plasticity (STP) was applied on excitatory and inhibitory synapses according to a well-established model [59]. Efferent synaptic connections from presynaptic units were modelled by the following system of ordinary differential equations:

$$\frac{du_j}{dt} = \frac{U - u_j}{\tau_f} + U(1 - u_j)\sum_k \delta(t - t_j^k) \tag{4}$$

$$\frac{dx_j}{dt} = \frac{1 - x_j}{\tau_d} - u_j x_j \sum_k \delta(t - t_j^k) \qquad (5)$$

where $\delta$ was the Dirac delta function, $t_j^k$ was the occurrence time of the $k$th spike of presynaptic unit $j$, $U \in [0, 1]$ represented the baseline release probability of $u_j$, and $x_j$ was the neurotransmitter availability. Presynaptic units in the network received intrinsic background input in the form of a spike train $s_j$, which was an independent homogeneous Poisson process for each presynaptic unit $j$ with rate $r$ of 10Hz [27]. For simplicity, the spatiotemporal pattern of the background noise remained constant during both evoked and spontaneous activity. In the absence of presynaptic spikes, unit $j$ was at a resting state, where $u_j$ and $x_j$ remained at baseline resting values $U$ and 1, respectively. When a presynaptic spike occurred, $u_j$ instantaneously increased $u_j \rightarrow u_j + U(1 - u_j)$, while $x_j$ instantaneously decreased $x_j \rightarrow (1 - u_j)x_j$. Between subsequent spikes, $u_j$ and $x_j$ recovered back to their baseline resting values $U$ and 1, respectively. Depending on the initial setup of parameters, $\tau_f$, $\tau_d$ and $U$, the phenomenological model of STP can mimic the effect of a depressing or a facilitating synapse [60]. Therefore, the mechanism of short-term depression and facilitation can be distinguished using a different parameter setup in the same governing equations. In our embodied network, excitatory and inhibitory synapses were subject to short-term depression, which is well-suited in shaping the temporal response properties of cortical neurons [59, 61, 62].

Spike-timing-dependent plasticity (STDP) was implemented using a pair-based, nearest-neighbour spike timing interaction [63]. At each time instance, the induction protocol evaluated the connection between presynaptic unit $j$ and postsynaptic unit $i$, and computed a time difference $\Delta t = t_i - t_j$ between the last spike emitted from each respective unit. In this way, the nearest time difference between spikes was continuously evaluated by the weight updating STDP rule described as:

$$\Delta J_{ij} = \begin{cases} \lambda_+ f_+(J_{ij}) e^{-\frac{\Delta t}{\tau_+}} & \text{if } \Delta t > 0 \\[2mm] -\lambda_- f_-(J_{ij}) e^{\frac{\Delta t}{\tau_-}} & \text{if } \Delta t \leq 0 \end{cases} \qquad (6)$$

$$f_+(J_{ij}) = (1 - J_{ij})^\mu \text{ and } f_-(J_{ij}) = \alpha(J_{ij})^\mu \qquad (7)$$

where $\lambda_\pm$, $0 < \lambda_\pm \ll 1$ were the learning rates $\lambda_+$ and $\lambda_-$ that scaled the magnitude of individual weight changes during long-term potentiation (LTP) and depression (LTD), respectively. The element $J_{ij}$ represented the strength and sign of the synapse from presynaptic unit $j$ to postsynaptic unit $i$. The updating functions $f_+(J_{ij})$ and $f_-(J_{ij})$ scaled weight changes and implemented LTP for a positive temporal difference $\Delta t > 0$, and LTD otherwise for a negative or null temporal difference $\Delta t \leq 0$ [64]. The parameter $\mu$ of the updating functions set boundary conditions on the changes in synaptic weights. Here, a non-zero $\mu$ was used in order to prevent runaway dynamics of synaptic weights. The parameter $\alpha$ denoted a possible asymmetry between the scales of LTD and LTP, where $\alpha < 1$ assigned a higher magnitude for LTP whereas $\alpha > 1$ favored LTD [64]. At several cortical synapses, the width of the plasticity window for LTD is considerably wider than the width of the plasticity window for LTP [63, 65]. Since LTD is widely expressed in the central nervous system [66], $\alpha$ was set greater than 1 and the temporally asymmetric STDP learning rule was evaluated with $\tau_- > \tau_+$. Parameter values of both STP and STDP were kept fixed throughout all robotic experiments (Fig 5).

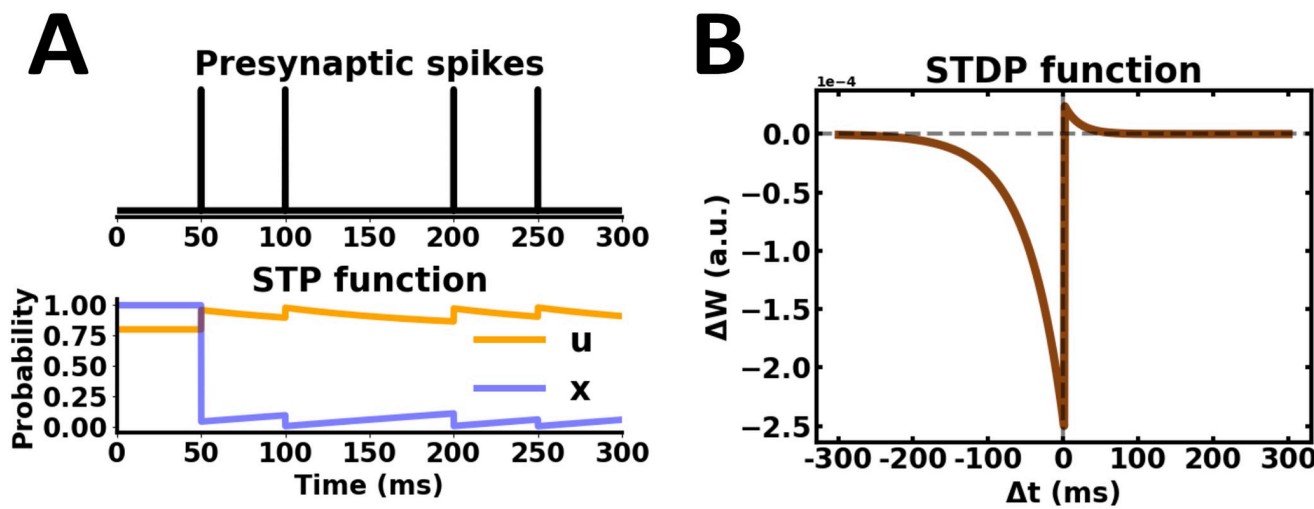

**Fig 5. Synaptic plasticity profile.** (A) STP dynamics from presynaptic spikes. Parameters: $\tau_f = 100$ ms, $\tau_d = 900$ ms, $U = 0.80$, $x = 1$. (B) Asymmetric STDP learning rule. Parameters: $\tau_- = 50$ ms, $\tau_+ = 20$ ms, $\lambda_- = 25e\text{-}5$, $\lambda_+ = 5e\text{-}5$, $\mu = 1$, $\alpha = 2$.

To combine the effects of STP and STDP, we first defined an instantaneous synaptic efficacy $u_j x_j$ with a time dependency due to STP (Eqs 4–5). From this, we further extended the instantaneous efficacy by adding $J_{ij}$ as an additional scaling factor [67–70], thus generating synaptic connections with efficacies that evolved according to:

$$G_i(t) = \sum_{j=1}^{N} J_{ij} \cdot u_j x_j \tag{8}$$

where $G_i$ represented the time-dependent total synaptic efficacy of postsynaptic unit $i$ mediated by the combined effects of STP and STDP. As a result, unit $i$ received at each time instance, a postsynaptic current $I_i^{syn}$ described by:

$$\frac{dI_i^{syn}}{dt} = -\frac{I_i^{syn}}{\tau_{syn}} + \sum_k G_i \delta(t - t_j^k) \tag{9}$$

where $t_j^k$ represented the $k$th spike emitted by presynaptic unit $j$ and $G_i$ was the peak amplitude of the elementary postsynaptic current triggered by the activation of presynaptic unit $j$. The Dirac delta function $\delta$ represented the occurrence of the presynaptic spike and $\tau_{syn}$ was the recovery time constant of the postsynaptic current. In particular, when the presynaptic unit $j$ emitted a spike, $I_i^{syn}$ was instantaneously raised according to $I_i^{syn} \rightarrow I_i^{syn} + G_i$, and then decayed exponentially back to 0 with a time constant $\tau_{syn}$ between subsequent spikes. For an overview of the computational model proposed, a flow chart is illustrated in Fig 6.

Initially, units in the network were connected according to a sparse matrix structure **C** (see Network architecture). If unit $j$ projected to unit $i$, then $C_{ij} = 1$. Otherwise, in the absence of a connection, $C_{ij} = 0$. As a result of the interaction between STP and STDP, the weight matrix **J** was engaged in structural plasticity such that novel synaptic contacts were formed over the course of the robotic experiment. Based on their difference in spike timing, uncoupled pairs of units $C_{ij} = 0$ formed a new connection $C_{ij} = 1$ for which their strength was scaled according to Eqs 6–7. In this way, the initial sparse network connectivity evolved towards a fully connected network (S1 Fig).

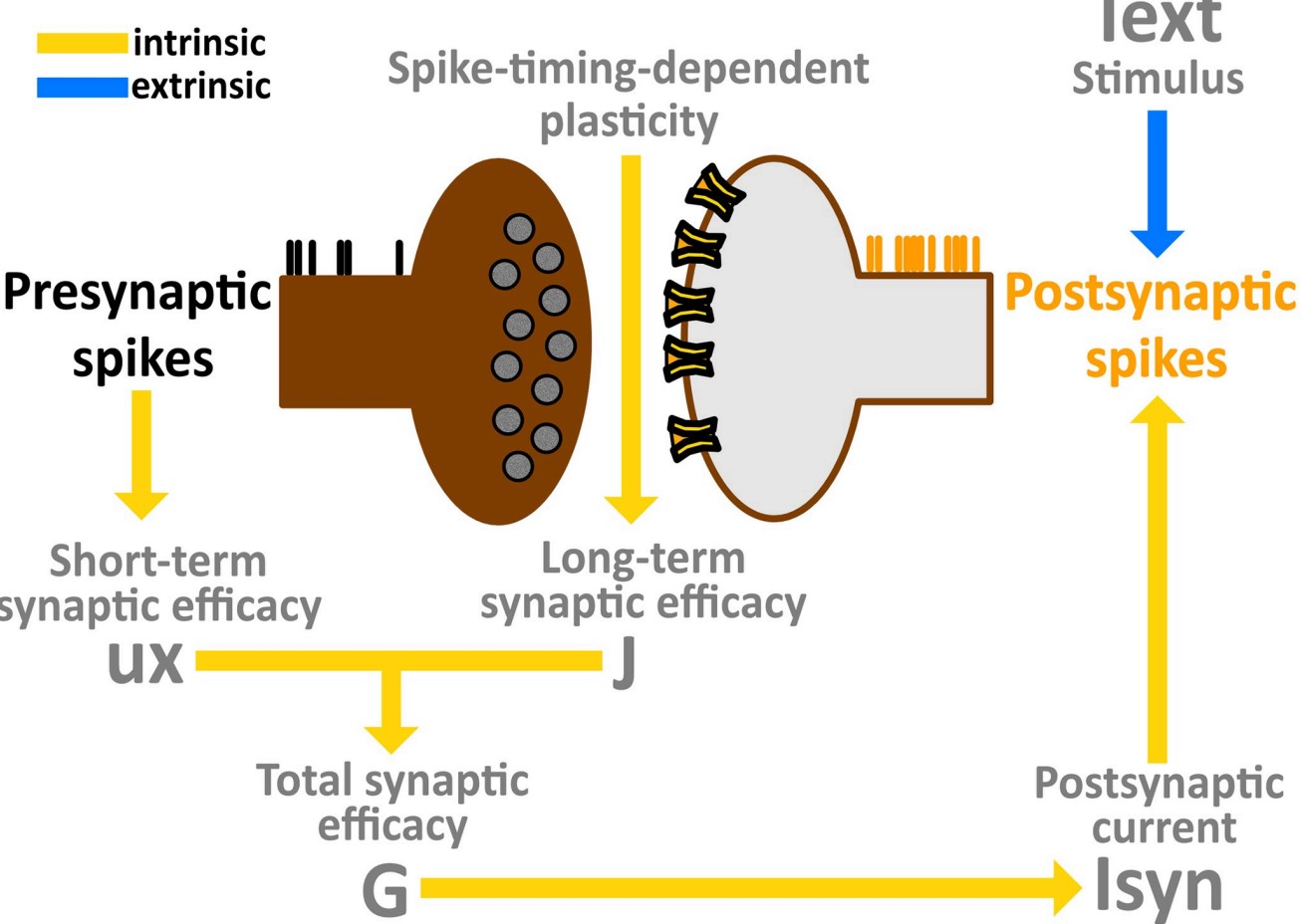

**Fig 6. Model flow chart.** Combination of short-term and long-term synaptic plasticity giving rise to a time-dependent total synaptic efficacy, which in turn generates a postsynaptic current that drives intrinsic network dynamics. Sensory afferents are characterized by an extrinsic input, directly affecting postsynaptic emission rates.

## Results

We report the results of six Robotic experiments. Each experiment is run via the Robotic platform, initiated using the initial Network architecture, and manipulated using the keyboard listening framework. Batches of evoked and spontaneous activity are alternated according to the Learning and recall procedure. Postsynaptic responses increase during evoked activity, and decrease during spontaneous activity (Fig 7). The rate of postsynaptic depolarization determines whether synaptic connections are potentiated or depressed. Overall, the robot exhibits (1) single learning and recall, (2) incremental learning and recall, (3) task-switching, (4) resistance to interference, (5) submission to interference, and (6) resistance to distraction. A video of each robotic experiment can be found in the (S1–S6 Videos).

### Experiment 1: Single learning and recall

In the first experiment, a single learning and recall procedure is performed (Fig 8). Here, external inputs are presented up to a point where the network reaches global stability. During evoked activity, neuronal responses follow the spatial profile of external inputs, exhibiting patterns of localized activity near the stimulus peaks (Fig 8A). Following stimulus offset, the

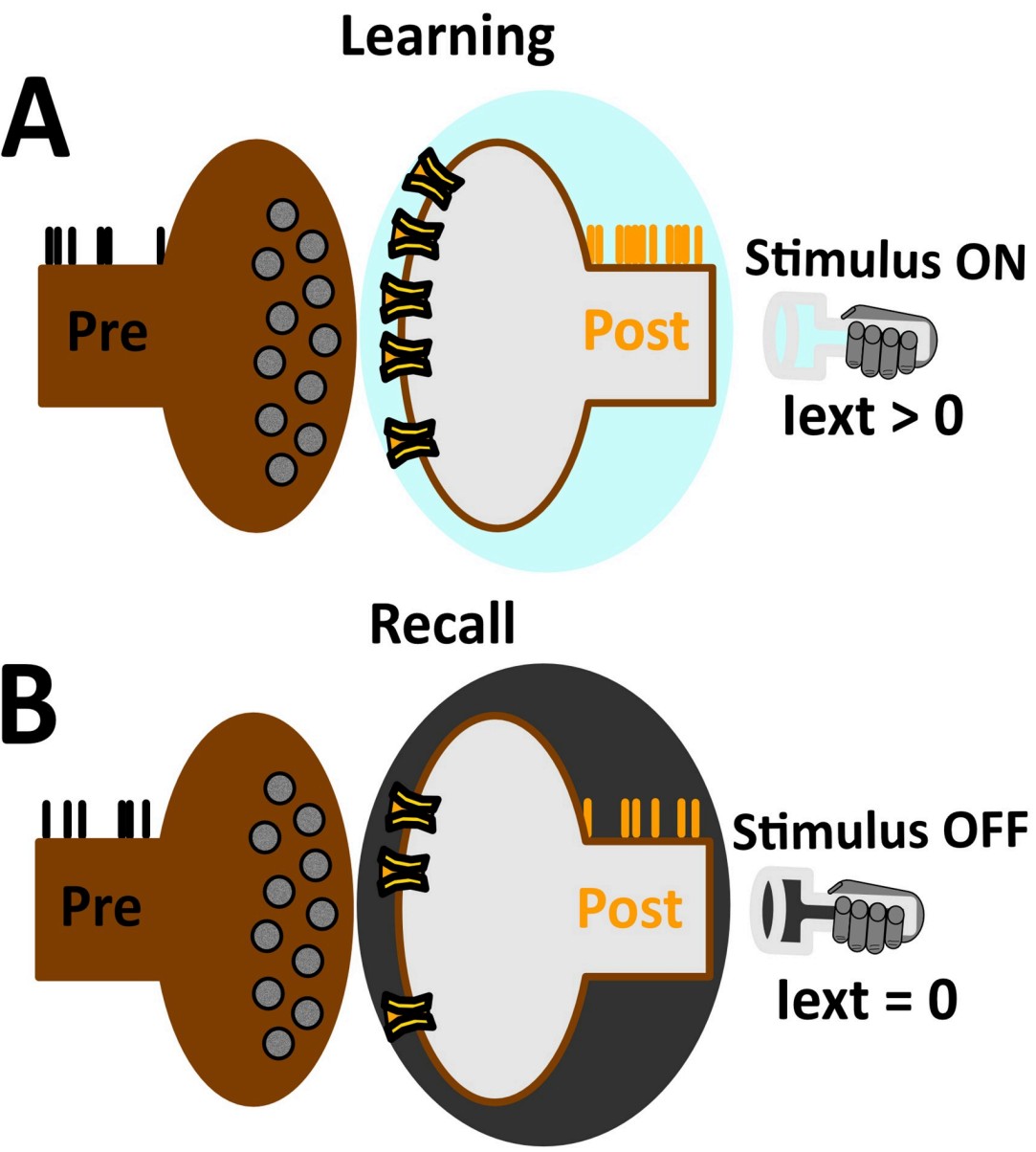

**Fig 7. Input-dependent and input-disengaged synaptic plasticity differentially modulate neuronal responses.** (A) When the stimulus is ON during learning, synaptic connections are strengthened by high postsynaptic emission rates. (B) When the stimulus is OFF during recall, synaptic connections are weakened by low postsynaptic emission rates. Presynaptic activity remains relatively constant.

network maintains local activity bumps (Fig 8A). These self-sustained reverberations are determined by the strength of synaptic efficacies established during evoked activity (Fig 8B). Noteworthily, synaptic efficacies can depend on the emission rate and spike timing of units they connect, both of which may jointly determine activity-dependent changes in synaptic plasticity [63]. Synapses connecting pairs of units most responsive near the stimulus peaks are strengthened. The preferential bias in local reverberating activity is reflected in the robot's wheel motor rotation speed (Fig 8C). As a result, movement trajectories during spontaneous activity are in the same direction as those observed during evoked activity (Fig 8D). Taken together, the network exhibits localized patterns of self-sustained activity with preferential

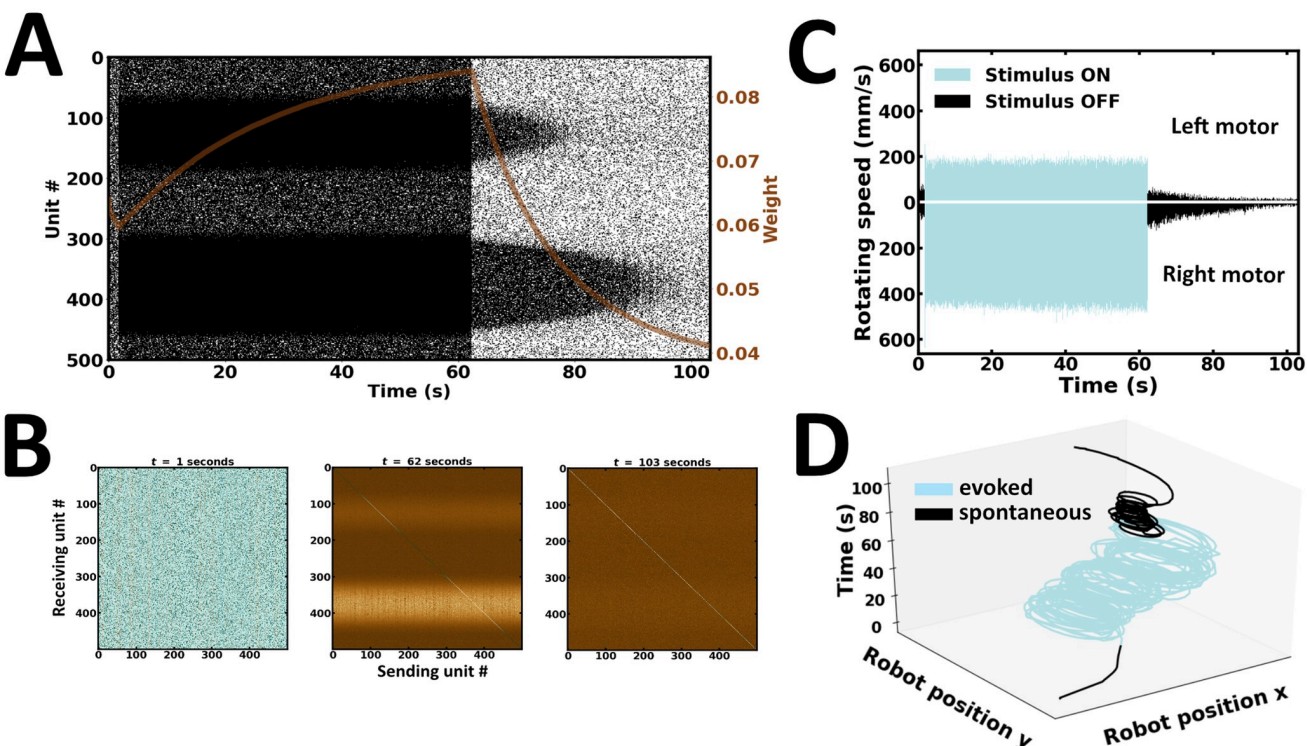

**Fig 8. Single learning and recall.** (A) Raster plot of network activity. Local persistent activity is preferentially biased towards a single subpopulation; brown curve, average synaptic efficacy. (B) Snapshot of synaptic weights at the end of evoked and spontaneous batches. (C) Rotating speed of left and right wheel motors as a function of time. (D) Motor trajectory of the robot in 3-D space during evoked and spontaneous batches.

responses that settle towards the position of the higher bump location, an observation directly manifested in the robot's motor trajectory (S1 Video).

## Experiment 2: Incremental learning and recall

In spite of persistent activity, networks with dynamic synapses are expected to display sensitivity to the temporal features of elicited stimuli, such as stimulus duration [34, 71–73]. Here, an incremental learning and recall procedure is introduced (Fig 9). Stimulus duration is progressively increased during successive presentations (Fig 9A). As a result, persistent activity progressively increases (Fig 9B). This gradual gain is associated with the strengthening of incoming synaptic connections (Fig 9C). As a result, differences in left and right wheel motor speed become progressively pronounced (Fig 9D). Consequently, the robot gradually refines its internal representation of sensory stimuli (Fig 9E). Noteworthily, autonomous motor trajectories gradually resemble more closely to the recall performance observed in Experiment 1. Taken together, the content of WM may be refined with longer stimulus presentations (S2 Video).

## Experiment 3: Task switching

In the third experiment, the robot is engaged in a task-switching procedure (Fig 10). Here, we examined whether the network is flexible enough to ride the wave of new input streams. To test this, stimulus intensity and duration are kept relatively constant (Fig 10A), whereas stimulus configuration is switched during successive evoked batches. As a result, the preferential

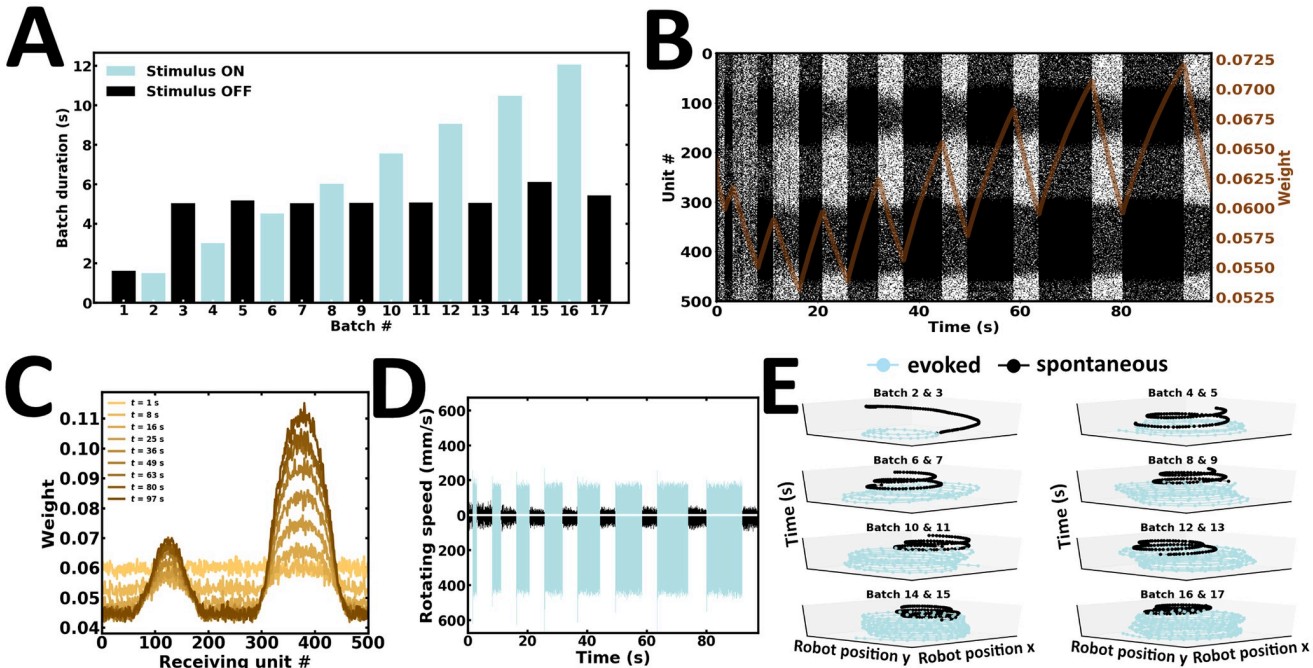

**Fig 9. Incremental learning and recall.** (A) Duration of evoked and spontaneous batches. (B) Raster plot of network activity. Persistent activity gain depends on stimulus duration; brown curve, average synaptic efficacy. (C) Evolution of average incoming synaptic weights at the end of spontaneous batches. (D) Rotating speed of left and right wheel motors as a function of time. (E) Motor trajectory of the robot in 3-D space during subsequent pairs of evoked and spontaneous batches.

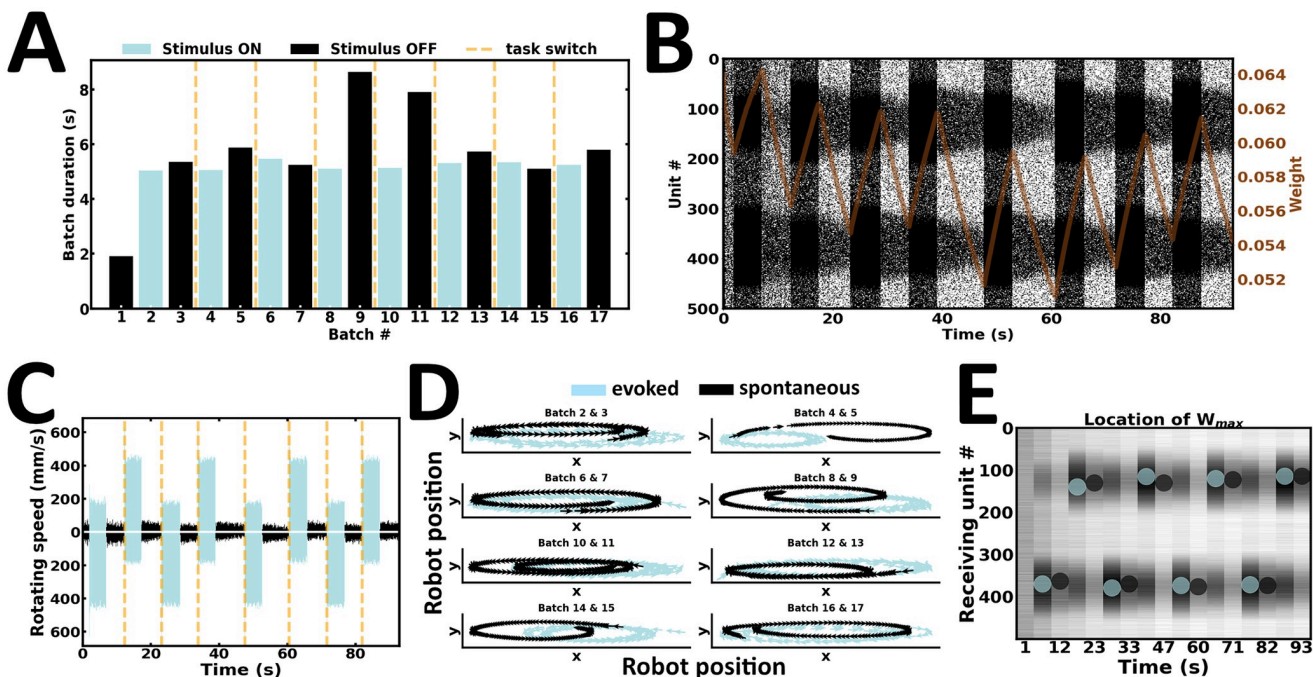

**Fig 10. Task switching.** (A) Duration of evoked and spontaneous batches. (B) Raster plot of network activity. Stimulus configuration shapes local persistent activity; brown curve, average synaptic efficacy. (C) Rotating speed of left and right wheel motors as a function of time. (D) Motor trajectory of the robot during subsequent pairs of evoked and spontaneous activity. (E) Spatial location of the highest average incoming synaptic weights; blue and black dot, evoked and spontaneous activity, respectively.

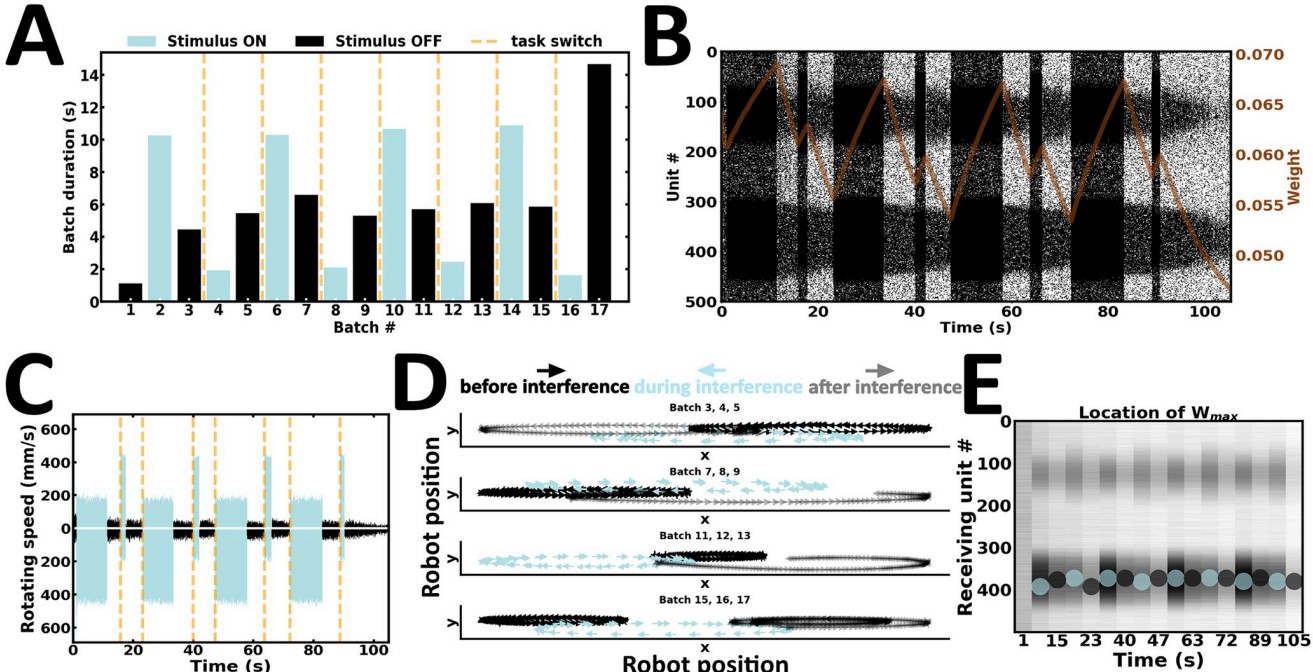

**Fig 11. Resisting interfering inputs.** (A) Duration of alternating batches of evoked and spontaneous activity. (B) Raster plot of network activity. Interfering inputs fail to disrupt cue-guided patterns of local persistent activity; brown curve, average synaptic efficacy. (C) Rotating speed of left and right wheel motors as a function of time. (D) Motor trajectory of the robot before, during and after the presentation of interfering inputs. (E) Spatial location of the highest average incoming synaptic weights; blue and black dot, evoked and spontaneous activity, respectively.

bias in local persistent activity is switched according to the latest input configuration (Fig 10B). Changes in local emission rates are directly reflected in the speed of wheel motors (Fig 10C). As a result, the robot actively maintains an internal representation of the latest stimulus presented (Fig 10D). Finally, the strength of synaptic weights near the higher stimulus peak changes accordingly (Fig 10E). Taken together, the content of WM may be overwritten by ongoing new input streams (S3 Video).

## Experiment 4: Resisting interfering inputs

In addition to switching tasks, animals can remain resistant to unforeseen interference from external stimuli [14, 46]. In the fourth experiment, the robot is tested against interference (Fig 11). Here, stimulus configurations are the same as those in Experiment 3. However, cue stimuli are presented for a longer duration than interfering inputs (Fig 11A). If the robot can resist interference, it should move in the same direction before and after interference. Fig 11B shows that cue-guided persistent activity is maintained following interference, suggesting that the network remains resistant to brief interfering stimuli. The same wheel motor remains higher before and after interference (Fig 11C). Consequently, the robot moves in the same direction before and after interference (Fig 11D) (S4 Video). Interestingly, this type of interference is not enough to alter the acquired structure of the synaptic weight matrix (Fig 11E). Taken together, our fourth experiment suggests that stimuli presented for a longer duration are more likely to be remembered than other stimuli. However, if different stimuli are successively presented for similar durations, the latest stimulus may overwrite the content of the previous stimulus, as observed in Experiment 3.

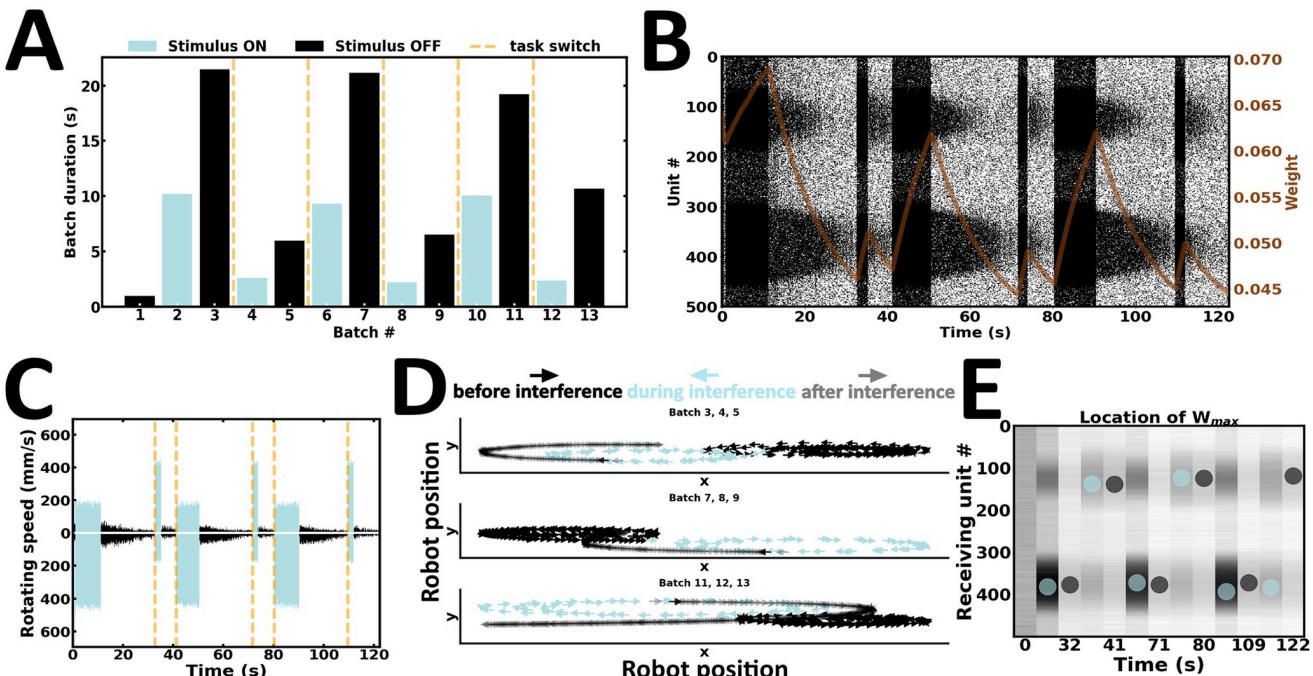

**Fig 12. Submitting to interfering inputs.** (A) Duration of alternating batches of evoked and spontaneous activity. (B) Raster plot of network activity. Interfering inputs disrupt cue-guided patterns of local persistent activity; brown curve, average synaptic efficacy. (C) Rotating speed of left and right wheel motors as a function of time. (D) Motor trajectory of the robot before, during and after the presentation of interfering inputs. (E) Spatial location of the highest average incoming synaptic weights; blue and black dot, evoked and spontaneous activity, respectively.

## Experiment 5: Submitting to interfering inputs

Despite the ability to resist interfering inputs in Experiment 4, cue stimuli were presented for a longer duration than interfering inputs. Arguably, the robot's movement trajectory being biased towards that of the stimulus presented longer seems fully expected. However, this interpretation remains agnostic about the contribution of delay duration, which introduces context under which the WM content either resists or submits to ongoing new input streams. To explain, if delay duration is extended, the WM content would be erased. After erasure, the robot is expected to remember the content of the latest stimulus presented, irrespective of stimulus duration. To test this hypothesis, the fifth experiment introduces a variant of Experiment 4 (Fig 12). Here, the duration of both stimuli is maintained, but delay activity is extended long enough to wash away the content of the cue (Fig 12A and 12B). Arguably, one would automate the paradigm using the exact same stimulus triggers as Experiment 4, and introduce an *n*-fold increase in delay duration. However, the latter approach is more suitable for disembodied networks, because multiple delay durations must be tested to find the minimum delay duration required to wash-away the content of the cue. To this end, multiple offline simulations runs would need to be performed, an approach our online proposal attempts to overcome. Hence, to move beyond offline experimental refinements, we turned to our online robotic implementation instead. Here, the robot accommodates interfering inputs by switching tasks, reminiscent of Experiment 3 (Fig 12C and 12D) (S5 Video). The strength of synaptic weights near the higher stimulus peak changes according to the ongoing wave of new input streams (Fig 12E). Taken together, our fifth experiment suggests that delay duration introduces context, under which the WM content either resists or submits to similar interferences.

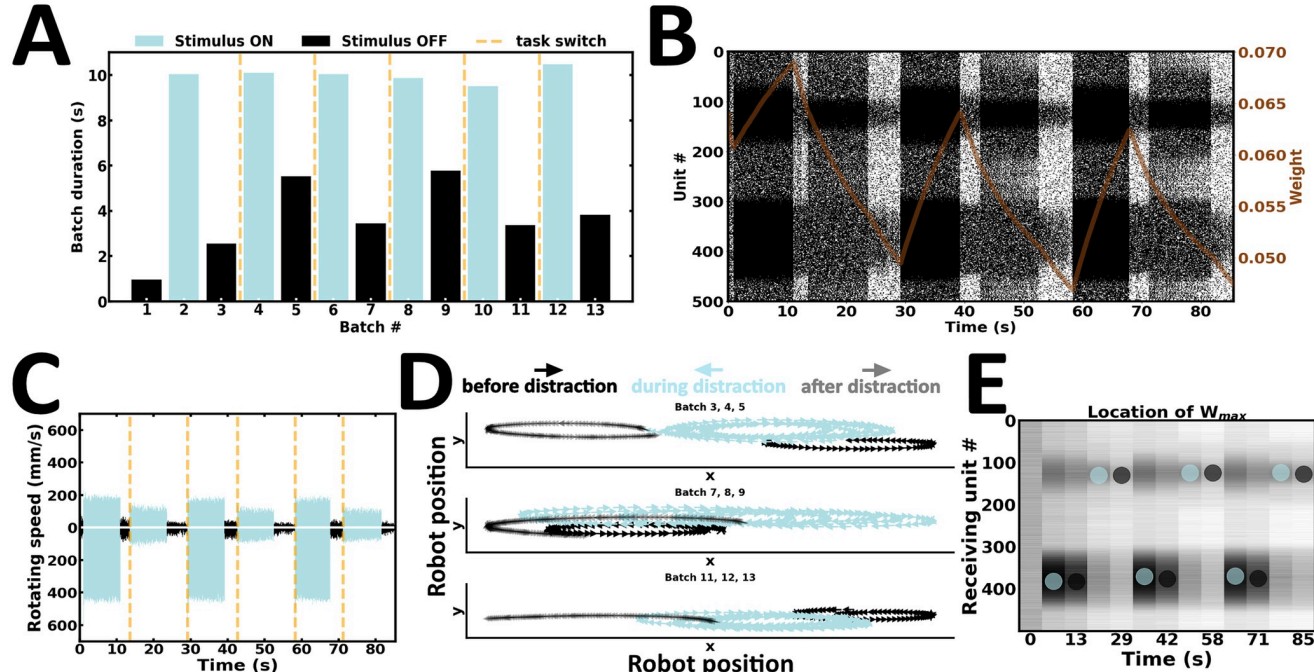

**Fig 13. Resisting distraction inputs.** (A) Duration of alternating batches of evoked and spontaneous activity. (B) Raster plot of network activity. Distraction inputs fail to disrupt cue-guided patterns of local persistent activity; brown curve, average synaptic efficacy. (C) Rotating speed of left and right wheel motors as a function of time. (D) Motor trajectory of the robot before, during and after the presentation of distraction inputs. (E) Spatial location of the highest average incoming synaptic weights; blue and black dot, evoked and spontaneous activity, respectively.

## Experiment 6: Resisting distraction inputs

In the sixth and final experiment, the robot is introduced to distraction inputs (Fig 13). Here, both cue and distractor durations are approximately the same (Fig 13A). However, distractor intensity is weaker than cue intensity. Cue-guided persistent activity is maintained following distraction inputs, suggesting that the WM content remains resistant to weaker stimuli (Fig 13B). The rotating speed of the same wheel motor remains higher before and after distraction (Fig 13C). Consequently, the robot executes movement trajectories in the same direction before and after distraction (Fig 13D) (S6 Video). Synaptic weights near stimulus peaks are continuously restructured during ongoing input streams (Fig 13E). Following distraction inputs, units controlling the left motors are connected by the strongest synaptic efficacies. In contrast, persistent activity is globally higher among units controlling the right motors. Therefore, local information may misrepresent population level activity under certain conditions. Taken together, our sixth and final experiment suggests that the content of WM may be resistant to distraction inputs, so long as the intensity of the latter is weaker than that of the cue stimulus.

## Discussion

### Relation to previous work

Working memory has been studied in networks of spiking neurons connected by synapses subject to both short-term and long-term plasticity [28]. Through mean field analysis, the authors showed that increased LTP should be adequately balanced by increased LTD to prevent uncontrollable excitation from destabilizing network activity. Moreover, they found that

the inclusion of short-term depression can widen the range of stable ongoing neural and synaptic dynamics, ensuring online learning is kept under control. This mean field approach was then supplemented by the work of [29], where the author studied how plastic synapses operate based on patterns of mean spike rates. The quantitative analysis showed that external stimuli can be embedded in the synaptic structure, controlled by the rate of postsynaptic depolarization above and below emission threshold. Furthermore, low LTP and LTD transition probabilities were shown to be instrumental for slow learning and forgetting to take place—a desirable feature in palimpsestic networks, where old memories are forgotten to make room for new ones [36–38]. In [30], the authors outlined general requirements for stimulus driven, unsupervised formation of WM states. Their mean field analysis suggested that Hebbian rate-dependent synaptic plasticity may serve in developing synaptic structure in initially unstructured networks. The role of short-term depression was highlighted, serving as a regulatory mechanism for preventing excessive potentiation from destabilizing stimulus-selective activity.

The theoretical studies of [27–30] made a handful of predictions for which our robotic implementation supports. First, our study shows how neural activity during ongoing input streams can lead to the formation of synaptic structures, which in turn support persistent activity. Second, the asymmetric STDP learning rule suggests that LTD is strong enough to compensate for excessive potentiation. Third, the regulatory role of short-term depression is supported by our online robotic implementation. Fourth, LTP and LTD learning rates are small enough to generate stable synaptic weight changes. Finally, transitions between LTP and LTD can be controlled by postsynaptic emission rates.

Short-term synaptic plasticity has been supported as a candidate mechanism for storing information in WM [24, 25]. Moreover, fast-expressing Hebbian synaptic plasticity may modify network connectivity momentarily enough to support information storage in WM [74–76]. Arguably, Hebbian forms of synaptic plasticity are incompatible with the flexible functionality of WM, because they induce long-lasting changes in synaptic connections that generally outlast the duration of persistent activity. However, persistent activity could be mediated by synapses that actively operate under multiple timescales [77], carefully orchestrating information maintenance in WM. Recently, *in vivo* recordings from dentate gyrus tracked the dynamics of synaptic changes after low interference WM training in freely behaving rats [78]. While animals were still performing the task, the authors hypothesized that prior WM contents could be forgotten so as to accurately recall the storage of new contents during forthcoming trials. Interestingly, WM erasure prior to good behavioural performance was associated with the physiological induction of LTD, suggesting its implication in the loss of prior irrelevant information for the adaptive benefit of storing new ones. Importantly, lack of WM erasure corresponded to more LTP-like phenomena, which impaired behavioural performance on subsequent trials [78]. Our robotic experiments are in qualitative agreement with these findings, suggesting that LTD may erase prior WM content during extended delay durations so that forthcoming input streams can be stored without proactive interference [79, 80]. Nevertheless, further experiments will be required to disentangle the role of fast-expressing Hebbian synaptic plasticity, and its potential involvement in WM storage [81–83].

In this study, sensory information was progressively lost during spontaneous activity [7]. In ANNs, this phenomenon is commonly known as catastrophic forgetting, where synaptic weights important for consolidating task A are changed to meet the requirements for task B [84–86]. A previous model of elastic weight consolidation (EWC) has attempted to overcome catastrophic forgetting by rendering a proportion of previous task-allocated synapses less plastic [84]. In this way, network activity reached stability over long timescales, without forgetting older tasks during sequential learning of multiple tasks. This approach may be instrumental for continual lifelong learning systems [87]. However, EWC weights are tailored for capturing

memory retention, and therefore the model does not consider a forgetting component, an active arbitrator of WM capacity.

In contrast to EWC weights, synapses in our network do not hold a specific mechanism designed to protect previous knowledge from being overwritten by learning of novel information [33, 84], nor are they imposed an active forgetting component. Rather, retention and forgetting are part of the ongoing dynamics, and there are no specialized resources administered to previous task-allocated synapses. The robotic experiments follow a continual learning scheme such that artificial stop-learning conditions are instead replaced by real-time user intervention. In light of this approach, our findings are aligned with previous theoretical studies, suggesting that the formation of WM may be subject to activity-dependent synaptic plasticity [88]. We show how this process can unfold in a robot embedded in an ecological context.

In this work, sensory information was treated as a distributed representation, where any individual neuron functions as part of a larger population whose combined activity underlies the agent's information processing capabilities [51]. Indeed, persistent activity is thought, in many cases, to carry distributed information—sensory representation is not a single neuron property, rather many neurons participate in the representation of an item loaded in memory [13]. Here, individual units responded in an idiosyncratic manner, but collectively they generated neuronal responses that provided a centralized control over each respective wheel motor of the robot. As such, wheel motors were controlled via the collective participation of distinct neuronal subpopulations. Motor trajectories were driven by distinct localized activity bumps. Indeed, these bump states are reminiscent to those observed in discrete slot models of multi-item WM capacity [17–19, 25, 50], where each item is stored in a bell-shaped activity bump [14–16, 19, 23, 25, 45–50].

Although the cognitive significance of persistent activity remains elusive, our work suggests that local subpopulations collectively extend their role in maintaining an active memory of structured, learned information about external stimuli [51]. Previous studies have shown that stable persistent activity could depend on a number of factors, namely the duration and intensity of the external input current applied, the synaptic weights, the level of intrinsic noise, as well as the size of the network [47]. Indeed, controlling the parameters that define the properties of the network provides a more powerful approach rather than simply changing the input to the network [45]. Although a complete assessment of these intricacies is beyond the scope of our robotic experiments, a future study could segregate the network from the physical robot, and investigate the sensitivity of the model to a wider range of parameters.

From a phenomenological standpoint, the fade-out bumps observed during the delay period imply the gradual disappearance of sensory representation [7]. In agreement with this finding, behavioural studies in human subjects have shown that WM precision can degrade with the duration of the delay period [89]. Analogously, the robot progressively fails to remember the direction of previously executed motor commands, which is reflected in the gradual disappearance of stimulus-selective activity towards baseline [26]. Noteworthily, persistent activity during the delay period is not only stimulus-dependent, but also likely to be mediated by time-varying fluctuations in neuronal activity which render the internal representation dynamic [90]. Indeed, WM experiments have shown a set of diverse neuronal response patterns during delay period activity [5, 6]. These individual response profiles range from sustained tonic activity to gradual increase/decrease in firing rate fluctuations. The presence of heterogeneity in the response of individual neurons likely provides a meaningful computational role for higher order function [39, 48, 91, 92]. Despite the presence of such diversity, our findings solely accounted for the progressive decrease in the neural response profiles [5, 6]. Nevertheless, persistent activity was maintained over a time scale of seconds [93, 94]. This observation has been a hallmark feature of the prefrontal cortex involved in WM, where the

regulation of internally-guided decisions can be executed in the absence of external stimuli [95]. From a neurorobotic standpoint, persistent activity is advantageous for artificial agents exposed to momentary hardware failure or sensor perturbation. Instead of relying purely on external stimuli, the agent internalizes recent events and carries them over towards a foreseeable future. In the absence of persistent activity, perceptual uncertainty would prevail, preventing an accurate execution of internally-guided behaviour.

## Limitations

In this study, executed motor trajectories aimed to capture the internal representation of sensory afferents stored in WM. However, using motor executions to characterize sensory representations is an abstraction that moves away from the kinds of representations memorized by animals during WM experiments. Therefore, the characterization of sensory representations via wheel motor executions creates a partial correspondence to the hypothesized contribution of WM maintenance proposed in experimental studies. Consequently, our robotic experiments create an imbalance in abstraction which may arguably lead to a loss of biological relevance [96]. It is important to note however that our work does not propose input-disengaged motor executions to be mediated by persistent activity. Rather, our study shows how the physical robot can be used as a companion approach for concretizing the representation of sensory stimuli in an embodied network. In particular, autonomous motor executions during spontaneous activity is used purely for the purpose of studying how the content of WM interacts with ongoing sensory afferents. Importantly, the inclusion of motor primitives turns out to be critical for providing behaviourally meaningful content to sensory primitives, because the maintenance of an ongoing movement holds a property in the action that accurately reflects the property of the sensory stimulus [97]. Beyond the sheer presence of neural activity and synaptic transmission, the motor primitives of the robot were used as a method for displaying persistent neural activity "in action". The robotic implementation was therefore used in order to ensure a one-to-one correspondence between network activity and executed motor trajectory. The extensive matching and complementarity between their properties constrains the scope of multifaceted interpretations in the face of non-stationary environments. Moreover, it provides an opportunity for their coordinated changes to ultimately result in the emergence of adaptive behaviour [98]. Overall, our robotic experiments aimed to exemplify the importance of embodiment as a tool for studying the formation of WM.

During the experiments, structural plasticity rendered an unrestricted establishment of novel synaptic connections within and between excitatory and inhibitory populations. In this context, excitatory units maintained their initial strict excitatory influences over the course of each experiment, because the multiplicative STDP rule linearly attenuates excitatory efficacies as they reach the boundaries [0, 1] [64]. As such, excitatory outgoing connections evolved within boundaries (S2 Fig—left column). In contrast, outgoing connections from inhibitory units switched their initial strict inhibitory influences by suddenly forming excitatory contacts with their novel postsynaptic targets, a biologically implausible scenario which led to the violation of Dale's law (S2 Fig—left column). To circumvent this issue, some networks have held fixed inhibitory efficacies [64, 99–101]. Others have attempted to obey Dale's principle by suddenly freezing synaptic connections if the synaptic update reversed the sign of excitatory or inhibitory influences [102]. In subsequent trainings, synapses attempting to change their signs were excluded and therefore prevented from exhibiting activity-dependent changes. Although the latter constraint is adequately tailored for the conservation of Dale's principle, an artificial stop-learning condition is imposed, which hinders the ability to learn new memories [85]. Nevertheless, model networks with positive and negative synaptic weights can be functionally

equivalent to those carrying positive weights only [103]. Finally, structural plasticity also led to the establishment of self-connections in our network, which were absent during network initialization (S2 Fig—right column).

In this work, inputs were represented as bimodal Gaussian mixtures (Sensory inputs). The motivation for using simplified sensory afferents partly stems from classic studies of visual WM, where subjects are introduced the oculomotor delayed response task [2, 6]. During the cue period of the task, stimuli are presented at various possible locations around a fixation point [104]. These stimuli may possess simple geometric shapes differing in luminance or color [7, 105]. Noteworthily, a substantial proportion of neurons in the dorsal region of the prefrontal cortex exhibit spatially tuned elevated persistent firing across the delay period [106]. However, the reverberating activity of prefrontal neurons can extend beyond the spatial domain, towards physical characteristics of objects (e.g. shape, color and intensity) and their respective identity [4, 107–113]. Although the bimodal mixture of Gaussians used in this study carries selectivity for spatial location and intensity, it remains agnostic to shape information, thus leaving out the multimodal characteristics of real world stimuli and the complex statistical structure they carry [114]. Nevertheless, it is possible to train the network on an arbitrary set of stimuli, such that delay activity is selective to object shape, as observed in prefrontal neurons [112]. An example is illustrated in the supplementary figure (S3 Fig).

## Practical implications

In this study, there are practical reasons for considering user keypresses instead of setting up timers to generate consistent experimental triggers. First, it is noteworthy to mention that the robot was tested in a confined spatial setting. Perhaps less obvious, devising a fully automated task in a restricted area runs the risk of having the robot run into collisions. Indeed, collisions may be avoided by running experiments in an unrestricted area or overcome by incorporating system feedback. However, our robot is designed to have low neglect tolerance, meaning it requires continuous supervisory control [44]. Agents with low neglect tolerance are not given the luxury of behaving in the context of automated task designs. Our goal is not to design automated paradigms, because the network is performed online and the robot is placed in an ecological context [115]. The study therefore focused on the interactive component between the user and the robot, such that the user received continuous behavioural feedback from the robot, and pressed a key at any moment, including when the agent ran the risk of collision. Real-time interventions as such created inconsistent experimental triggers.

The keyboard listening framework developed here is suitable for online learning of rapidly varying input streams, a phenomenon known as concept drift [32]. In particular, the framework is designed for choosing a specific input configuration from a repertoire of configurations, such that it can introduce novel motor trajectories via rapid and flexible task-switching. Future studies could use our keyboard listening framework and expand the repertoire of input configurations. This approach has been proposed as an important design principle in developmental robotics [43, 116]. Since the programmer who develops the keyboard listening framework does not know what tasks the future users will teach the robot, the framework gives the freedom to choose a sequence of tasks that are unknown at the time of programming [116]. Hence, in an attempt to further explore how the model behaves with disparate problems in the same network, programmers may design a wider range of stimulus duration, intensity, specificity and bump location. To this end, we expect that the computations resulting from a wider range of input configurations and durations may not only reproduce the behavioural results presented here, but also further extend the range of possible behavioural trajectories. For example, introducing a bimodal Gaussian mixture with localized amplitude peaks of the same

intensity should remove the preferential bias from the persistent activity bumps observed here, making local reverberating activity between the two subpopulations indistinguishable. Consequently, the robot would autonomously navigate forward following the removal of the "unbiased" stimuli. This new behavioural trajectory would directly result from the coupled interaction between neural activity and motor output, both of which are under the influence of ongoing new input streams presented to the same underlying network [98].

Robotic behaviour can be used a method for making an inference about the underlying network computation. Adaptive behaviour is directly observable, because the online operation is tailored for piece-by-piece accommodation of ongoing input streams. In this way, new items can be actively loaded in WM and therefore the agent can be trained to maintain novel sensory representations on-the-fly [116]. Moreover, the keypresses used here minimize the efforts required to make arbitrary changes in the elicited stimuli [117]. Interactions as such provide external observers some time to interpret the behaviour of the robot, and form a real-time hypothesis of its forthcoming behaviour. Under the umbrella of this interaction, the robot can actively augment, and be shaped by the user. Taken together, our study proposes that traditional WM protocols may be complemented by placing biological agents in an ecological context, under which the content of WM interacts with ongoing input streams.

## Conclusion

In this work, we have devised a network of spiking neurons with dynamic synapses. As a companion approach to the computational model, the network was embedded in a physical robot. The artificial agent was used as a source of exploration under which neuronal computation was directly linked to motor output—serving as an illustration of WM capacity under non-stationary input streams. This methodological approach was accompanied by a keyboard listening framework, which helped us delineate how the content of WM is (1) refined, (2) overwritten or (3) resisted by changes in the duration and configuration of incoming stimuli. The model we propose is capable of continuous online learning, and the embodied network adapts to changing environmental contexts, all of which are done so under the basis of local information reliance. We show how these local changes can be accounted for by short-term and long-term changes in synaptic plasticity. Although we believe that carefully designed experiments will continue to lay the foundation for our understanding of higher-order function, we hope that embodied networks will add a supporting layer to the valuable theoretical insights gained from computational models of pure simulation.

## Supporting information

**S1 Fig. Temporal evolution of structural plasticity.**
(TIF)

**S2 Fig. Temporal evolution of synaptic weights.**
(TIF)

**S3 Fig. Persistent activity based on object shape and location.**
(TIF)

**S1 Video. Robotic experiment 1.** Single learning and recall.
(MP4)

**S2 Video. Robotic experiment 2.** Incremental learning and recall.
(MP4)

**S3 Video. Robotic experiment 3.** Task-switching.
(MP4)

**S4 Video. Robotic experiment 4.** Resistance to interference.
(MP4)

**S5 Video. Robotic experiment 5.** Submission to interference.
(MP4)

**S6 Video. Robotic experiment 6.** Resistance to distraction.
(MP4)

# Acknowledgments

We thank Professor Georg Northoff for providing invaluable feedback on an earlier version of this manuscript.

# Author Contributions

**Conceptualization:** Nareg Berberian, Sylvain Chartier.

**Formal analysis:** Nareg Berberian, Matt Ross.

**Funding acquisition:** Sylvain Chartier.

**Investigation:** Nareg Berberian.

**Methodology:** Nareg Berberian, Matt Ross, Sylvain Chartier.

**Resources:** Sylvain Chartier.

**Supervision:** Sylvain Chartier.

**Visualization:** Nareg Berberian, Sylvain Chartier.

**Writing – original draft:** Nareg Berberian, Matt Ross, Sylvain Chartier.

**Writing – review & editing:** Nareg Berberian, Matt Ross, Sylvain Chartier.

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
