## [Decision Letter · Decision Letter 0]

19 Aug 2020

PONE-D-20-18770

Embodied working memory during ongoing input streams

PLOS ONE

Dear Dr. Berberian,

Thank you for submitting your manuscript to PLOS ONE. After careful consideration, we feel that it has merit but does not fully meet PLOS ONE’s publication criteria as it currently stands. Therefore, we invite you to submit a revised version of the manuscript that addresses the points raised during the review process.

As you'll see from the reviews, both reviewers found multiple points where the paper was unclear.

We look forward to receiving your revised manuscript.

Kind regards,

William W Lytton, MD

Academic Editor

PLOS ONE

Journal Requirements:

Reviewers' comments:

Reviewer's Responses to Questions

**Comments to the Author**

1. Is the manuscript technically sound, and do the data support the conclusions?

Reviewer #1: Yes

Reviewer #2: Partly

2. Has the statistical analysis been performed appropriately and rigorously? 

Reviewer #1: No

Reviewer #2: No

3. Have the authors made all data underlying the findings in their manuscript fully available?

Reviewer #1: No

Reviewer #2: Yes

4. Is the manuscript presented in an intelligible fashion and written in standard English?

Reviewer #1: Yes

Reviewer #2: Yes

5. Review Comments to the Author

Reviewer #1: This report is mostly a 'methods' paper describing a system for studying working memory. The authors claim that the most unique feature is that stimulus afferents to the modeled working memory are continuous. Their model does not employ gates are predetermined connectivity strengths to work, rather it evolves entirely based on a combination of short and longer term plasticity changes.

The biggest problem I have with the paper is the simplicity of sensory afferent. It is modeled as a bimodal mixture of Gaussians. I would like to have seen a discussion of how this compares to stimuli in the 'real' world The authors seem to address this tangentially by indicating that future studies would include more complex stimuli, but do not discuss the limitations of the simple stimuli used.

'Neurons' explicitly do not connect to themselves in the model. But it is not clear whether as connections are modified during learning self-connections do occur. Moreover, it appears that connections can evolve from being excitatory to inhibitory over the course of learning. The discussion of whether this is a violation of Dale's principle is opaque and it is not clear whether the authors have already addressed this issue.

The role of noise and randomness in the model is not explicitly addressed; perhaps this is why there was no need for repeated simulations and statistical analyses of the simulations.

The raster plots that are presented for each of the figures, the plots of weight changes over time, and the 3-dimensional tracing of robot movements probably capture all of the data of the paper, but downloadable data files would be more consistent with the spirit of question 3 above.

I like this paper and believe it should be published. I would have preferred if the authors added paragraphs to the introduction and discussion giving concrete examples about how this work differs from other models of working memory.

Reviewer #2: In this paper the authors present an SNN hooked up to a Vector robot and run it through a series of different input sequences. The resulting activity of the network is discussed.

I found the paper subjects to be interesting (SNNs and embodied robotics), but the overarching goals and relevance of the robotic agent in the experiments unclear. The experiments are framed as the network accomplishing various memory tasks, such as intermittent learning and recall, but I think explicit goals need to be laid out for each before this is an appropriate presentation. I'm not sure that the labelling of the tasks is consistent with the literature. For example, it is my understanding that one shot learning is usually a paired stimulus-response learning task, and less the case where the network continues to output the same signal after the stimulus is removed.

At the beginning, the artificial agent is presented as a tool for studying embodied working memory; I think this needs to be further developed in this paper to support this claim. With no sensory feedback from the robot, I'm not sure I see the benefit of using it.

I think it also important that the language be simplified throughout for better clarity.

My main concern, however, is with the experiments themselves. I think it's important to specifically laying out criteria for successful completion for each. I also have the following questions.

Experiment 1: What makes this unique from any network that converges to a set of learned weights? Do all such networks display one-shot learning and recall?

Experiment 2: Is this the same as Experiment 1, but the network was simulated without input at several points before it converged? As opposed to Experiment 1 where input was provided until convergence and then it was simulated without input.

Experiment 3: Are you expecting to see the rotating speed of the different motors increase over time as the tasks are learned? It appears in Fig 8-C that the activity that happens right after instantiation is consistent throughout, which suggests to me that no sustained learning is happening.

Experiment 4: This experiment is framed as 'overcoming intervening stimuli', but that's not reflective of the task. The task is applying one stimulus for a longer amount of time than a different stimulus. The framing of this an 'an ability of the network to overcome distractions' feels like an overstatement. The result of the network activity being biased towards that of the stimulus presented longer seems fully expected.

Overall, I think there is the core of some interesting work looking at SNNs and robotics, but I believe there is some substantial rework required with the experiments. If the goal is to show the use of an artificial agent there should be some form of system feedback incorporated. Without incorporating feedback I could also just assign different outputs to different behaviours in a consistent way without any need for an agent.

For these reasons I am recommending rework and resubmit.

Larger comments:

The language is complex, I believe the information can be conveyed with much simpler phrasing.

I believe the experiment paradigm should be fully automated as opposed to human run (pushing the buttons). What is the motivation for not setting up timers to generate consistent experimental triggers?

I'd be interested to see a short mention of supporting biological evidence for working memory storing information through synaptic changes instead of in, say, population level activity.

It's not clear to me how the Experiments build on one another.

Nits:

'Reverberatory activity' is a phrase I haven't seen before. Is this not 'recurrent activity'? If 'reverberatory' is not established language in a field I suggest changing to 'recurrent' for clarity.

For all non-3D plots, can you put time on the x axis?

Line 55: SNNs require continuous modification of synaptic connections

- What is this based on? Are you suggesting it is not possible to engineer functional SNNs without plasticity?

- The citation shows how continuous learning _can_ be used in SNNs but does not appear to suggest they _require_ it.

Line 84: Why are self connections not permitted? A summary of the motivation for these rules should be added.

Line 99-100:

- What is the relevancy of these references?

- Is there a section of the book that is of interest?

- Presumably the alternative to teleoperation is embedded control? This paragraph says both you're not reliant on physical sensors, but also you instantly connect incoming stimuli and the robot. Do you mean incoming stimuli as in user keypresses?

- Why would reliance on physical sensors vastly limit the fluidity of information exchange during the presence of ongoing input streams? Please rework for clarity.

Line 103: Should set timers or triggers in the code instead of involving a human for better reproduce ability.

Fig 1: Vector is an adorable robot.

Line 117-118: There are just the two peaks, correct? highest -> higher, lowest -> lower

Line 116: What's the motivation for using bimodal activations and the specific offset between the two Gaussians?

Table 2: Remove and include parameters in Fig 2 caption.

Table 3: Remove, text description is sufficient, table isn't adding anything.

Line 171: Learning and recall -> Learning and recall phases

Line 189: I believe 'elicited' is the wrong word, 'the elicited stimuli' -> 'the stimuli'

Line 201: How were the learning rates chosen?

Table 4: A small figure showing the profile with these parameters in the caption would be more useful to the reader.

Table 5: Same comment.

Line 291: I would move this first paragraph to the Learning / Recall paradigm discussion to help the reader frame what the actual experiments being run are while reading through methods.

Line 342: This suggests something else would determine the activity aside from connection weights, what would that be? The temporal dynamics of the neuron model?

Fig 8-D: axes incorrectly labelled

6. PLOS authors have the option to publish the peer review history of their article (what does this mean?). If published, this will include your full peer review and any attached files.

Reviewer #1: No

Reviewer #2: No

---

## [Author Response · Author response to Decision Letter 0]

8 Nov 2020

Reviewer #1: This report is mostly a 'methods' paper describing a system for studying working memory. The authors claim that the most unique feature is that stimulus afferents to the modeled working memory are continuous. Their model does not employ gates are predetermined connectivity strengths to work, rather it evolves entirely based on a combination of short and longer term plasticity changes.

The biggest problem I have with the paper is the simplicity of sensory afferent. It is modeled as a bimodal mixture of Gaussians. I would like to have seen a discussion of how this compares to stimuli in the 'real' world The authors seem to address this tangentially by indicating that future studies would include more complex stimuli, but do not discuss the limitations of the simple stimuli used.

Author response: We thank the reviewer for this recommendation. The following paragraph has been added in subsection entitled: Limitations (lines 626-641):

In this work, inputs were represented as bimodal Gaussian mixtures (see Sensory inputs). The motivation for using simplified sensory afferents partly stems from classic studies of visual WM, where subjects are introduced the oculomotor delayed response task (Funahashi et al., 1989; Takeda & Funahashi, 2002). During the cue period of the task, stimuli are presented at various possible locations around a fixation point (Constantinidis et al., 2018). These stimuli may possess simple geometric shapes differing in luminance or color (Constantinidis et al., 2001; Hoshi et al., 1998). Noteworthily, a substantial proportion of neurons in the dorsal region of the prefrontal cortex exhibit spatially tuned elevated persistent firing across the delay period (Meyer et al., 2011). However, the reverberating activity of prefrontal neurons can extend beyond the spatial domain, towards physical characteristics of objects (e.g. shape, color and intensity) and their respective identity (Freedman et al., 2001; Miller et al., 1996; O Scalaidhe, 1997; Ó Scalaidhe et al., 1999; G. Rainer et al., 1998; Gregor Rainer & Miller, 2000; Rao, 1997; Wilson et al., 1993). Although the bimodal mixture of Gaussians used in this study carries selectivity for spatial location and intensity, it remains agnostic to shape information, thus leaving out the multimodal characteristics of real world stimuli and the complex statistical structure they carry (Brader et al., 2007). Nevertheless, it is possible to train the network on an arbitrary set of stimuli, such that delay activity is selective to object shape, as observed in prefrontal neurons (Rao et al., 1997). An example is illustrated in the supplementary figure (S3 Fig).

'Neurons' explicitly do not connect to themselves in the model. But it is not clear whether as connections are modified during learning self-connections do occur. Moreover, it appears that connections can evolve from being excitatory to inhibitory over the course of learning. The discussion of whether this is a violation of Dale's principle is opaque and it is not clear whether the authors have already addressed this issue.

Author response: We thank the reviewer for raising these two points to our attention. Discussion about the violation of Dale’s law and the evolution of self-connections has been expanded in subsection entitled: Limitations (lines 606-625):

During the experiments, structural plasticity rendered an unrestricted establishment of novel synaptic connections within and between excitatory and inhibitory populations. In this context, excitatory units maintained their initial strict excitatory influences over the course of each experiment, because the multiplicative STDP rule linearly attenuates excitatory efficacies as they reach the boundaries [0, 1] (Gütig et al., 2003). As such, excitatory outgoing connections evolved within boundaries (S2 Fig – left column). In contrast, outgoing connections from inhibitory units switched their initial strict inhibitory influences by suddenly forming excitatory contacts with their novel postsynaptic targets, a biologically implausible scenario which led to the violation of Dale's law (S2 Fig – left column). To circumvent this issue, some networks have held fixed inhibitory efficacies (Clopath et al., 2010; Gütig et al., 2003; Hosaka et al., 2008; Sadeh et al., 2015). Others have attempted to obey Dale's principle by suddenly freezing synaptic connections if the synaptic update reversed the sign of excitatory or inhibitory influences (Kim & Chow, 2018). In subsequent trainings, synapses attempting to change their signs were excluded and therefore prevented from exhibiting activity-dependent changes. Although the latter constraint is adequately tailored for the conservation of Dale's principle, an artificial stop-learning condition is imposed, which hinders the ability to learn new memories (Knoblauch et al., 2014). Nevertheless, model networks with positive and negative synaptic weights can be functionally equivalent to those carrying positive weights only (Parisien et al., 2008). Finally, structural plasticity also led to the establishment of self-connections in our network, which were absent during network initialization (S2 Fig – right column).

The role of noise and randomness in the model is not explicitly addressed; perhaps this is why there was no need for repeated simulations and statistical analyses of the simulations.

Author response: Indeed, our results rest upon the robotic implementation, which is used as a direct instrument for assessing the accuracy of network performance. 

The raster plots that are presented for each of the figures, the plots of weight changes over time, and the 3-dimensional tracing of robot movements probably capture all of the data of the paper, but downloadable data files would be more consistent with the spirit of question 3 above.

Author response: Data files of all experiments will be provided during resubmission. 

I like this paper and believe it should be published. I would have preferred if the authors added paragraphs to the introduction and discussion giving concrete examples about how this work differs from other models of working memory.

Author response: We thank the reviewer for the positive assessment of our work. New paragraphs have been added in the Introduction and Discussion sections, comparing our work to previous WM models. 

Introduction (lines 19-25): 

…These formal models, among others, form recurrently connected networks, which typically maintain selective elevated persistent activity through local excitatory recurrent connections with global feedback inhibition (Koulakov et al., 2002). In this formalism, sustained firing may be achieved by carefully fine-tuning the strength and structure of recurrent circuitry (Barak & Tsodyks, 2014). As such, these functional networks can maintained, in memory, the spatial location of target stimuli, forming what is commonly known as persistent activity bumps (i.e. bump attractors) (Koulakov et al., 2002).

Introduction (lines 26-32):

Persistent activity in recurrent networks has been supported by short-term synaptic plasticity, where synaptic strength is rapidly regulated by recent historical activity within the network (Itskov et al., 2011; Schneegans & Bays, 2018; Seeholzer et al., 2019). In the presence of such rapid changes in synaptic dynamics, neuronal activity can drift over time (Schneegans & Bays, 2018; Seeholzer et al., 2019), or remain centered at the initial bump location (Itskov et al., 2011). Despite their overarching support from empirical studies (Barbosa et al., 2020; Edin, Klingberg, Johansson, McNab, Tegnér, et al., 2009), WM models holding the persistent memory hypothesis have for long been using preassigned protocols of fixed stimulus presentation… 

Introduction (lines 43-52):

In contrast to displaying persistent activity by recurrent interactions, here we instead devise a feedforward network of spiking neurons (Goldman, 2009), subject to both short-term and long-term synaptic plasticity (D. Amit, 1997; Paolo Del Giudice et al., 2003; Fusi, 2002; P. D. Giudice & Mattia, 2001). By considering activity-dependent Hebbian plasticity as a complementary mechanism for generating persistent activity (Manohar et al., 2019), we examine how the content of WM interacts with the intricacies of ongoing input streams. Spiking networks are well-known for carrying out continuous online operations in non-stationary environments (Lobo et al., 2020). In this context, synaptic connections can be continuously modified (Zenke et al., 2017), without fine-tuning their strength and structure. This ongoing process of modification may result in momentary network restructuring so as to accommodate an evolving environment (Lobo et al., 2020)…

Relation to previous work (lines 460-478): 

Working memory has been studied in networks of spiking neurons connected by synapses subject to both short-term and long-term plasticity (P. Del Giudice & Mattia, 2001). Through mean field analysis, the authors showed that increased LTP should be adequately balanced by increased LTD to prevent uncontrollable excitation from destabilizing network activity. Moreover, they found that the inclusion of short-term depression can widen the range of stable ongoing neural and synaptic dynamics, ensuring online learning is kept under control. This mean field approach was then supplemented by the work of (Fusi, 2002), where the author studied how plastic synapses operate based on patterns of mean spike rates. The quantitative analysis showed that external stimuli can be embedded in the synaptic structure, controlled by the rate of postsynaptic depolarization above and below emission threshold. Furthermore, low LTP and LTD transition probabilities were shown to be instrumental for slow learning and forgetting to take place -- a desirable feature in palimpsestic networks, where old memories are forgotten to make room for new ones (D. J. Amit & Fusi, 1994; Nadal et al., 1986; Parisi, 1986). In (Paolo Del Giudice et al., 2003), the authors outlined general requirements for stimulus driven, unsupervised formation of WM states. Their mean field analysis suggested that Hebbian rate-dependent synaptic plasticity may serve in developing synaptic structure in initially unstructured networks. The role of short-term depression was highlighted, serving as a regulatory mechanism for preventing excessive potentiation from destabilizing stimulus-selective activity.

Relation to previous work (lines 479-487):

The theoretical studies of (D. Amit, 1997; Paolo Del Giudice et al., 2003; Fusi, 2002; P. Del Giudice & Mattia, 2001) made a handful of predictions for which our robotic implementation supports. First, our study shows how neural activity during ongoing input streams can lead to the formation of synaptic structures, which in turn support persistent activity. Second, the asymmetric STDP learning rule suggests that LTD is strong enough to compensate for excessive potentiation. Third, the regulatory role of short-term depression is supported by our online robotic implementation. Fourth, LTP and LTD learning rates are small enough to generate stable synaptic weight changes. Finally, transitions between LTP and LTD can be controlled by postsynaptic emission rates.

Relation to previous work (lines 488-496):

Short-term synaptic plasticity has been supported as a candidate mechanism for storing information in WM (Barbosa et al., 2020b; Edin, Klingberg, Johansson, McNab, Tegnér, et al., 2009). Moreover, fast-expressing Hebbian synaptic plasticity may modify network connectivity momentarily enough to support information storage in WM (Fiebig et al., 2020; Fiebig & Lansner, 2017; Sandberg et al., 2003b). Arguably, Hebbian forms of synaptic plasticity are incompatible with the flexible functionality of WM, because they induce long-lasting changes in synaptic connections that generally outlast the duration of persistent activity. However, persistent activity could be mediated by synapses that actively operate under multiple timescales (Duarte et al., 2017), carefully orchestrating information maintenance in WM…

Reviewer #2: In this paper the authors present an SNN hooked up to a Vector robot and run it through a series of different input sequences. The resulting activity of the network is discussed.

I found the paper subjects to be interesting (SNNs and embodied robotics), but the overarching goals and relevance of the robotic agent in the experiments unclear. The experiments are framed as the network accomplishing various memory tasks, such as intermittent learning and recall, but I think explicit goals need to be laid out for each before this is an appropriate presentation. I'm not sure that the labelling of the tasks is consistent with the literature. For example, it is my understanding that one shot learning is usually a paired stimulus-response learning task, and less the case where the network continues to output the same signal after the stimulus is removed.

At the beginning, the artificial agent is presented as a tool for studying embodied working memory; I think this needs to be further developed in this paper to support this claim. With no sensory feedback from the robot, I'm not sure I see the benefit of using it.

Author response: We are thankful to the reviewer in seeing room for improvement. A new subsection has been added in the revised manuscript entitled: Robotic experiments – laying out the explicit goal of each experiment and the criteria for their successful completion (please see author response to Experiments 1-4). Please also note the two additional experiments (5-6) included in the revised manuscript. Finally, task labels have changed where needed so as to remain consistent with the literature. 

I think it also important that the language be simplified throughout for better clarity.

Author response: We appreciate this suggestion and have revised the manuscript throughout to simplify the writing. 

My main concern, however, is with the experiments themselves. I think it's important to specifically laying out criteria for successful completion for each. I also have the following questions.

Experiment 1: What makes this unique from any network that converges to a set of learned weights? Do all such networks display one-shot learning and recall?

Author response: 

Robotic experiments (lines 211-220): 

In the first experiment, a single learning and recall procedure is introduced. Since network activity is unavailable to the user, the behaviour of the robot is monitored to gain insight into network convergence. Here, the criteria for successful task completion is to observe consistent movement trajectories from the robot. This approach differs from disembodied networks imprinted with a minimum stopping rule for weight convergence, because robotic behaviour is the marker of network stability, not preassigned internal triggers. Using movement trajectories as top-down evidence of network convergence, the user can then remove the stimulus and observe the recall performance of the robot. This observation is used as a stepping stone for introducing an incremental paradigm.

Experiment 2: Is this the same as Experiment 1, but the network was simulated without input at several points before it converged? As opposed to Experiment 1 where input was provided until convergence and then it was simulated without input.

Author response: The reviewer makes an accurate comparison between Experiments 1 & 2. However, it noteworthy to highlight the robotic implications of Experiment 2. 

Robotic experiments (lines 221-229): 

In the second experiment, an incremental learning and recall procedure is presented. This experiment encompasses the notion of behavioural shaping (Chernova & L. Thomaz, 2014). Here, shaping refers to an incremental process where the user provides feedback to an agent so as to improve approximations of a target behaviour (Knox et al., 2013, 2009). The user observes the behaviour of the robot in response to different feedback signals (e.g. stimulus duration), and makes the necessary adjustments on-the-fly. In this way, robotic behaviour is progressively fine-tuned, moving the agent closer to the desired behaviour (Saksida et al., 1997). This incremental process towards a single target behaviour begs the question, however, whether the robot can move beyond single task demands, by transitioning between multiple tasks.

Experiment 3: Are you expecting to see the rotating speed of the different motors increase over time as the tasks are learned? It appears in Fig 8-C that the activity that happens right after instantiation is consistent throughout, which suggests to me that no sustained learning is happening.

Author response: We thank the reviewer for highlighting this observation. 

Robotic experiments (lines 230-235): 

In the third experiment, a task-switching procedure is introduced. The robot is shown two interchanging inputs. The criteria for success is the expression of flexible behaviour. Here, sustained learning is not a necessary requirement for successful task completion, because task-switching washes away the memory of the previous task. The robot must adapt to change at every step of the way, accommodating user desirability. Nevertheless, is the robot always deemed to be accurately aligned with user judgements?

Experiment 4: This experiment is framed as 'overcoming intervening stimuli', but that's not reflective of the task. The task is applying one stimulus for a longer amount of time than a different stimulus. The framing of this an 'an ability of the network to overcome distractions' feels like an overstatement. The result of the network activity being biased towards that of the stimulus presented longer seems fully expected.

Author response: 

Robotic experiments (lines 236-242): 

In the fourth experiment, a task interference procedure is introduced. Here, the robot is presented with a long cue stimulus, and tested against a brief interfering stimulus. To successfully accomplish this experiment, the robot must keep performing movement trajectories in the same direction as those observed during the cue period, even after the user interferes by imposing movements in the opposite direction. Nevertheless, if the robot resists interfering inputs, is it because they are presented for a shorter duration or because the content of the cue stimulus is still maintained in WM?

Above, the reviewer raises an important point: despite the ability to resist interfering inputs, cue stimuli are presented for a longer duration than interfering inputs. Arguably, the robot's movement trajectory being biased towards that of the stimulus presented longer seems fully expected. 

It is noteworthy to mention however that this interpretation remains agnostic about the contribution of delay duration, which introduces context under which the WM content either resists or submits to ongoing new input streams. To explain, if delay duration is extended, the WM content would be erased. After erasure, the robot is expected to remember the content of the latest stimulus presented, irrespective of stimulus duration. 

Experiment 5: submitting to interfering inputs (lines 426-436):

…To test this hypothesis, the fifth experiment introduces a variant of Experiment 4 (Fig 12). Here, delay duration is extended long enough to wash-away the content of the cue stimulus. Arguably, one would automate the paradigm using the exact same stimulus triggers as Experiment 4, and introduce an n-fold increase in delay duration. However, the latter approach is more suitable for disembodied networks, because multiple delay durations must be tested to find the minimum delay duration required to wash-away the content of the cue. To this end, multiple offline simulations runs would need to be performed, an approach our online proposal attempts to overcome. Hence, to move beyond offline experimental refinements, we turned to our online robotic implementation instead…

Robotic experiments (lines 243-250): 

In the fifth experiment, a variant of the task interference procedure is presented. Here, delay duration is extended long enough to wash-away the content of the cue stimulus. When the robot forgets the cue stimulus, its movement trajectory provides the user with enough evidence to present the interfering stimulus. To successfully perform this experiment, brief interfering stimuli should be enough to overwrite the content of cue stimuli. Otherwise, the previous experiment would suggest that the robot resisted interfering inputs simply because they were presented for a shorter duration, and not because the content of cue stimuli were maintained in WM…

Indeed, results from the fifth experiment show that the robot accommodates interfering inputs by switching tasks, reminiscent of Experiment 3. Delay duration therefore introduces context under which the WM content either resists or submits to similar interferences. Taken together, the fifth experiment suggests that resistance to shorter interfering inputs is not fully expected, if the contribution of delay duration is considered. Nevertheless, we agree with the reviewer, in that framing Experiment 4 as “overcoming intervening stimuli” is an overstatement. We have instead reframed the latter as “resisting interfering stimuli”.

Robotic experiments (lines 250-251): 

…Nevertheless, is the robot capable of resisting distractions that last as long as the cue stimulus? 

To address this question, we ran a sixth and final experiment.

Robotic experiments (lines 252-260):

In the sixth and final experiment, the robot is tested against distractor inputs. Here, the duration of both cue and distractor inputs are similar. However, distractor intensity is three times weaker than cue intensity. The magnitude of distractor intensity is based on two requirements, namely (1) to impose distracting movements in a direction opposite to those imposed during the cue period and (2) to navigate within the confined spatial environment. To successfully accomplish this experiment, the robot must move in the same direction before and after distraction, despite movements in the opposite direction during distraction. Taken together, the behaviour of the robot, not just the network activity, is a requirement for the successful completion of our experiments.

Experiment 6: resisting distraction inputs (lines 455-457):

…Taken together, our sixth and final experiment suggests that the content of WM may be resistant to distraction inputs, so long as the intensity of the latter is weaker than that of the cue stimulus.

Overall, I think there is the core of some interesting work looking at SNNs and robotics, but I believe there is some substantial rework required with the experiments. If the goal is to show the use of an artificial agent there should be some form of system feedback incorporated. Without incorporating feedback I could also just assign different outputs to different behaviours in a consistent way without any need for an agent.

For these reasons I am recommending rework and resubmit.

Author response: We appreciate the reviewer in seeing the value of studying physically instantiated networks of spiking neurons. Undoubtedly, system feedback would add an additional layer of complexity to the experiments that deserves the pursuit in its own right, but moves beyond the scope of our current user-robot interaction scheme. Nevertheless, we acknowledge the limitations of our study, and justify the reasoning behind the use of our robotic implementation. 

Larger comments:

The language is complex, I believe the information can be conveyed with much simpler phrasing.

Author response: We appreciate this comment and have revised the manuscript throughout to simplify the writing.

I believe the experiment paradigm should be fully automated as opposed to human run (pushing the buttons). What is the motivation for not setting up timers to generate consistent experimental triggers?

Author response: We thank the reviewer for this question. A new paragraph has been added under the subsection entitled: Practical implications; justifying the motivation for not setting up timers to generate consistent experimental triggers (lines 643-656):

In this study, there are practical reasons for considering user keypresses instead of setting up timers to generate consistent experimental triggers. First, it is noteworthy to mention that the robot was tested in a confined spatial setting. Perhaps less obvious, devising a fully automated task in a restricted area runs the risk of having the robot run into collisions. Indeed, collisions may be avoided by running experiments in an unrestricted area or overcome by incorporating system feedback. However, our robot is designed to have low neglect tolerance, meaning it requires continuous supervisory control (Goodrich & Schultz, 2007). Agents with low neglect tolerance are not given the luxury of behaving in the context of automated task designs. Our goal is not to design automated paradigms, because the network is performed online and the robot is placed in an ecological context (Amershi et al., 2014). The study therefore focused on the interactive component between the user and the robot, such that the user received continuous behavioural feedback from the robot, and pressed a key at any moment, including when the agent ran the risk of collision. Real-time interventions as such created inconsistent experimental triggers.

I'd be interested to see a short mention of supporting biological evidence for working memory storing information through synaptic changes instead of in, say, population level activity. 

Author response: We appreciate the reviewer in mentioning this point. We have added the following paragraph in subsection entitled: Relation to previous work (lines 488-509):

Short-term synaptic plasticity has been supported as a candidate mechanism for storing information in WM (Barbosa et al., 2020; Edin et al., 2009). Moreover, fast-expressing Hebbian synaptic plasticity may modify network connectivity momentarily enough to support information storage in WM (Fiebig et al., 2020; Fiebig & Lansner, 2017; Sandberg et al., 2003). Arguably, Hebbian forms of synaptic plasticity are incompatible with the flexible functionality of WM, because they induce long-lasting changes in synaptic connections that generally outlast the duration of persistent activity. However, persistent activity could be mediated by synapses that actively operate under multiple timescales (Duarte et al., 2017), carefully orchestrating information maintenance in WM. Recently, in vivo recordings from dentate gyrus tracked the dynamics of synaptic changes after low interference WM training in freely behaving rats (Missaire et al., 2020). While animals were still performing the task, the authors hypothesized that prior WM contents could be forgotten so as to accurately recall the storage of new contents during forthcoming trials. Interestingly, WM erasure prior to good behavioural performance was associated with the physiological induction of LTD, suggesting its implication in the loss of prior irrelevant information for the adaptive benefit of storing new ones. Importantly, lack of WM erasure corresponded to more LTP-like phenomena, which impaired behavioural performance on subsequent trials (Missaire et al., 2020). Our robotic experiments are in qualitative agreement with these findings, suggesting that LTD may erase prior WM content during extended delay durations so that forthcoming input streams can be stored without proactive interference (Underwood, 1957; Wixted, 2004). Nevertheless, further experiments will be required to disentangle the role of fast-expressing Hebbian synaptic plasticity, and its potential involvement in WM storage (Erickson et al., 2010; Park et al., 2014; Pradier et al., 2018). 

It's not clear to me how the Experiments build on one another.

Author response: Indeed, experiments do not build on one another per se. Rather, each experiment provides an entry point for the next one. We have changed the original statement, and clarified the reasoning behind the transition between experiments. Please see subsection entitled: Robotic experiments.

Nits:

'Reverberatory activity' is a phrase I haven't seen before. Is this not 'recurrent activity'? If 'reverberatory' is not established language in a field I suggest changing to 'recurrent' for clarity.

Author response: Indeed, 'reverberatory activity' is not as commonly used in the working memory literature anymore, but see (D. J. Amit, 1995; Sandberg et al., 2003). We have instead changed it to 'reverberating activity', a more common terminology for describing self-sustained persistent activity.

For all non-3D plots, can you put time on the x axis?

Author response: Ok, we have now put time on the x axis for all non-3D plots. 

Line 55: SNNs require continuous modification of synaptic connections

- What is this based on? Are you suggesting it is not possible to engineer functional SNNs without plasticity?

Author response: Plasticity is not a necessary requirement for engineering functional SNNs (Abbott et al., 2016). Our statement is referring to SNNs subject to online learning, where network structure can change on-the-fly to accommodate new input streams. 

- The citation shows how continuous learning _can_ be used in SNNs but does not appear to suggest they _require_ it.

Author response: Thank you for highlighting the overstatement. We have changed the sentence in the revised manuscript (lines 49-51): 

…In this context, the efficacy of their synaptic connections can be continuously modified (Zenke et al., 2017), without the necessity of carefully fine-tuning synaptic weights…

Line 84: Why are self connections not permitted? A summary of the motivation for these rules should be added.

Author response: Ok, we now mention why self-connections were not permitted at network initialization, and provide a summary of the motivation for the rules in subsection entitled: Network architecture (lines 86-97):

First, inhibitory neurons occupy approximately 20% of the population of cortical neurons, whereas the great majority of remaining neurons are excitatory (Gabbott & Somogyi, 1986). As such, 80% of units in the embodied network were randomly chosen to be excitatory, whereas the remaining 20% were inhibitory. Following Dale's law, a given excitatory/inhibitory unit only exhibited excitatory/inhibitory efferent connections, respectively. Second, the number of afferent synapses to a single neuron in cortex is limited, forming clusters of sparsely connected networks with roughly 10-20% probability of synaptic contact (Braitenberg & Schüz, 1998). To this end, initial network connectivity was sparse, with only 20% of all possible connections present (chosen randomly among all possible connections). At network initialization, self-connections were not permitted because although not uncommon, they are rare in vivo, in comparison to their prominence in dissociated cell cultures (Bekkers, 2003).

Line 99-100:

- What is the relevancy of these references?

Author response: In (WENG, 2004), pressure sensors are used for teaching humanoid robots by direct push action and force. These onboard sensors are directly linked to corresponding actuators. The reference deserves to be highlighted because in contrast to our approach, their study suggests that fluid communication can be maintained using onboard touch sensors. 

In (Goodrich & Schultz, 2007), the authors discuss design principles for framing the problem of human-robot interaction. They outline various communication methods and describe constituent measures of efficient interaction. The latter include (1) mental workload and (2) time requirement for conveying user desirability, among others. The communication method in our study (i.e. user keypresses) reduces both efficiency measures (see response below regarding fluidity of information exchange). For this reason, we believe the reference to be relevant.

- Is there a section of the book that is of interest? 

Author response: Section 6.3 in (WENG, 2004) and section 4.2 in (Goodrich & Schultz, 2007) are of interest.

- Presumably the alternative to teleoperation is embedded control? This paragraph says both you're not reliant on physical sensors, but also you instantly connect incoming stimuli and the robot. Do you mean incoming stimuli as in user keypresses?

Author response: We thank the reviewer for the alternative terminology: embedded control. Indeed, incoming stimuli are referred to as user keypresses. We clarify these points in subsection entitled: Robotic platform (lines 106-110):

The computational resources of the robot were remote, performed on a laptop computer. Incoming stimuli were controlled via user keypresses from the laptop computer, and the communication was mediated via wireless network. This embedded control design was intended so as to avoid using Vector's onboard tactile sensors.

- Why would reliance on physical sensors vastly limit the fluidity of information exchange during the presence of ongoing input streams? Please rework for clarity.

Author response: Ok, we have elaborated on this point in subsection entitled: Robotic platform (lines 110-119):

To explain, the robot continuously moved during the experiments. Therefore, reliance on tactile sensors would demand the user to continuously follow the robot throughout the experiment. Moreover, sensor reading is noisy and therefore some tactile triggers would go unnoticed from Vector failing to read them as valid touch. These would vastly limit the fluidity of information exchange, because they would place high workload demands on the user (but see (WENG, 2004)). Consequently, the time required for conveying user desirability would increase. For these reasons, we have instead devised a keyboard listening framework. The protocol was designed to ensure an efficient and instant medium of communication between the user and the robot (Goodrich & Schultz, 2007).

Line 103: Should set timers or triggers in the code instead of involving a human for better reproduce ability.

Author response: We appreciate the reviewer for this comment. As a reminder, we kindly ask to refer to comments above, where we justify our motivation for not setting up consistent timers. Nevertheless, we highly value open access and reproducibility. An automated version of the code, along with data from all experiments will be made available on the corresponding author’s GitHub repository. 

Fig 1: Vector is an adorable robot.

Author response: Thank you very much, it is our intent in making it intelligent yet humble. 

Line 117-118: There are just the two peaks, correct? highest -> higher, lowest -> lower

Author response: Yes this is correct. We have made these changes in the revised manuscript.

Line 116: What's the motivation for using bimodal activations and the specific offset between the two Gaussians?

Author response: The motivation has now been highlighted in subsection entitled: Sensory inputs (lines 139-150):

There are three main reasons why bimodal activations were used, including the specified offset between the two Gaussians. First, previous models of spatial WM have used the Gaussian for characterizing their target stimulus. Inputs range from a single target to multiple targets (Bouchacourt & Buschman, 2019; Compte, 2000; Edin, Klingberg, Johansson, McNab, Tegner, et al., 2009; Eliasmith, 2005; Hansel & Mato, 2013; Itskov et al., 2011; Kilpatrick, 2018; Laing & Chow, 2001; Renart et al., 2003; Sandamirskaya, 2014; Seeholzer et al., 2019; Tanaka, 2002). Second, Vector is equipped with two wheel motors. The agent was therefore an ideal candidate for supplying its differential steering system with bimodal activations. The specific offset between the Gaussians induced differences in wheel motor speed, which in turn resulted in rotating movements with diameters small enough for the robot to accommodate the confined spatial setting. Finally, the offset between the two Gaussians has experimental grounding, inspired from a variant of the classic oculomotor delayed response task (Constantinidis et al., 2001). In these experiments, animals are shown two simultaneously presented stimuli, with differences in luminance (i.e. contrast ratio). 

Table 2: Remove and include parameters in Fig 2 caption.

Author response: Ok, Table 2 has been removed and parameters have instead been included in Fig 2 caption. 

Table 3: Remove, text description is sufficient, table isn't adding anything.

Author response: Ok, Table 3 has been removed. 

Line 171: Learning and recall -> Learning and recall phases

Author response: Thank you, 'phases' has been added on line 179. 

Line 189: I believe 'elicited' is the wrong word, 'the elicited stimuli' -> 'the stimuli'

Author response: Ok, we have removed the word 'elicited' from 'the elicited stimuli'.

Line 201: How were the learning rates chosen?

Author response: Learning rates were chosen to be small enough for slow and steady adjustment of synaptic weights. 

Table 4: A small figure showing the profile with these parameters in the caption would be more useful to the reader.

Author response: Ok, the table has been replaced by a new figure (Fig 4) with aEIF model parameters included in the caption. 

Table 5: Same comment.

Author response: Ok, the table has been replaced by a new figure (Fig 5) with STP and STDP parameters included in the caption.

Line 291: I would move this first paragraph to the Learning / Recall paradigm discussion to help the reader frame what the actual experiments being run are while reading through methods.

Author response: We thank the reviewer for the suggestion. That paragraph is now immersed in the new subsection entitled: Robotic experiments; guiding the reader through the actual experiments. 

Line 342: This suggests something else would determine the activity aside from connection weights, what would that be? The temporal dynamics of the neuron model?

Author response: Indeed, the temporal dynamics of the neuron model may have an impact on the activity, because synaptic efficacies can depend on both emission rate and spiking timing of units they connect, both of which may jointly determine activity-dependent changes in synaptic plasticity (Sjöström et al., 2001). This point is now added in the revised manuscript (lines 367-369). 

Fig 8-D: axes incorrectly labelled

Author response: Thank you, Fig 8-D is now Fig 10-D in the revised manuscript, and its axes have been modified.

---

## [Decision Letter · Decision Letter 1]

17 Dec 2020

Embodied working memory during ongoing input streams

PONE-D-20-18770R1

Dear Dr. Berberian,

We’re pleased to inform you that your manuscript has been judged scientifically suitable for publication and will be formally accepted for publication once it meets all outstanding technical requirements.

Kind regards,

William W Lytton, MD

Academic Editor

PLOS ONE

Additional Editor Comments (optional):

Reviewers' comments:

Reviewer's Responses to Questions

**Comments to the Author**

1. If the authors have adequately addressed your comments raised in a previous round of review and you feel that this manuscript is now acceptable for publication, you may indicate that here to bypass the “Comments to the Author” section, enter your conflict of interest statement in the “Confidential to Editor” section, and submit your "Accept" recommendation.

Reviewer #1: All comments have been addressed

Reviewer #2: All comments have been addressed

2. Is the manuscript technically sound, and do the data support the conclusions?

Reviewer #1: Yes

Reviewer #2: Yes

3. Has the statistical analysis been performed appropriately and rigorously? 

Reviewer #1: N/A

Reviewer #2: N/A

4. Have the authors made all data underlying the findings in their manuscript fully available?

Reviewer #1: Yes

Reviewer #2: Yes

5. Is the manuscript presented in an intelligible fashion and written in standard English?

Reviewer #1: Yes

Reviewer #2: Yes

6. Review Comments to the Author

Reviewer #1: (No Response)

Reviewer #2: (No Response)

7. PLOS authors have the option to publish the peer review history of their article (what does this mean?). If published, this will include your full peer review and any attached files.

Reviewer #1: No

Reviewer #2: **Yes: **Travis DeWolf

---

## [Editor Report · Acceptance letter]

22 Dec 2020

PONE-D-20-18770R1 

Embodied working memory during ongoing input streams 

Dear Dr. Berberian:

I'm pleased to inform you that your manuscript has been deemed suitable for publication in PLOS ONE. Congratulations! Your manuscript is now with our production department. 

Kind regards, 

on behalf of

Dr. William W Lytton 

Academic Editor

PLOS ONE